# The impact of hydrological model structure on the simulation of extreme runoff events

Gijs van Kempen[1], Karin van der Wiel[2], and Lieke Anna Melsen[1]

[1]Hydrology and Quantitative Water Management, Wageningen University, Wageningen, the Netherlands
[2]Royal Netherlands Meteorological Institute (KNMI), De Bilt, the Netherlands

**Correspondence:** lieke.melsen@wur.nl

**Abstract.** Hydrological extremes affect societies and ecosystems around the world in many ways, stressing the need to make reliable predictions using hydrological models. However, several different hydrological models can be selected to simulate extreme events. A difference in hydrological model structure results in a spread in the simulation of extreme runoff events. We investigated the impact of different model structures on the magnitude and timing of simulated extreme high- and low-flow events, by combining two state-of-the-art approaches; a modular modelling framework (FUSE) and large ensemble meteorological simulations. This combination of methods created the opportunity to isolate the impact of specific hydrological process formulations at long return periods without relying on statistical models. We showed that the impact of hydrological model structure was larger for the simulation of low-flow compared to high-flow events and varied between the four evaluated climate zones. In cold and temperate climate zones, the magnitude and timing of extreme runoff events were significantly affected by different parameter sets and hydrological process formulations, such as evaporation. In the arid and tropical climate zones, the impact of hydrological model structures on extreme runoff events was smaller. This novel combination of approaches provided insights into the importance of specific hydrological process formulations in different climate zones, which can support adequate model selection for the simulation of extreme runoff events.

## 1 Introduction

Extreme high and low-flow events, often referred to as floods and droughts, respectively, have high natural, societal and economic impacts. On the global scale, fatalities and economic losses related to high-flow events have increased dramatically over the past decades (Di Baldassarre et al., 2010; Winsemius et al., 2016), among others due to an increase of settlements in flood prone regions. The impacts of low-flow events can be recognised in amongst others the water supply, crop production, and the hydro-power sectors (Van Loon, 2015). To mitigate the societal impact of hydrological extremes, knowledge of the processes leading to these extreme events is vital. Hydrological modelling is one of the main tools in this quest for knowledge, but comes with uncertainties. Here we aim to investigate the impact of hydrological model structure on the magnitude and timing of simulated extreme runoff events.

Hydrological mitigation efforts often relate to the return period of the extreme event, a measure that describes the 'extreme-ness' of the events. It is a traditional method to relate the magnitude of an event to the probability of occurrence of the event (Gumbel, 1941; Salas and Obeysekera, 2014), based on which decision makers can define their policy. Frequency analysis of extremes aims at estimating runoff levels corresponding to certain return periods (Laio et al., 2009). However, the limited length of available observational hydrological records means we rely on statistical models to estimate return periods most of the time (Meigh et al., 1997; Michele and Rosso, 2001; Smith et al., 2015; Sousa et al., 2011), e.g. by fitting a Generalized Extreme Value (GEV) distribution.

Despite the wide application of GEV analysis to relate runoff to return periods, there are some important caveats to this method. The statistical models are particularly used for extrapolation - to estimate the probability of yet unobserved extremes. As such, the projected hydrological extremes are highly sensitive to small changes in the parameters of statistical models (Engeland et al., 2004; Smith et al., 2015; Klemeš, 2000), leading to distributions that might substantially change when a single data point is added (see e.g. Brauer et al., 2011). Furthermore, the physical processes leading to extrapolated extreme events can not be investigated. A recent alternative to extreme value statistics models, proposed for example by Van der Wiel et al. (2019), is to use large ensemble model simulations: a climate model simulates long time series of meteorological conditions, and with a hydrological model this is translated to runoff, resulting in a long time series that does not require extrapolation for the investigation of events of longer return periods.

In this approach, hydrological models are employed to translate meteorological time series into hydrological time series, from which relevant events can be selected and investigated. Uncertainty is, however, also inevitable in model simulations (Oreskes et al., 1994). In hydrological modelling, different sources of uncertainty can be distinguished, for instance data uncertainty, parameter uncertainty and model structural uncertainty (Ajami et al., 2007). Data uncertainty can be related to random or systematic errors in the model forcing. Parameter uncertainty can be caused by sub-optimal identification of parameter values or equifinality, and model structural uncertainty relates to incomplete or biased model structures (Butts et al., 2004). It is important to gain insight in the uncertainty of environmental models and to communicate these insights to decision makers (Liu and Gupta, 2007), especially in the perspective of extreme events that give rise to policy making.

Data uncertainty and parameter uncertainty can be quantified by a combination of error propagation and sampling (Li et al., 2010; McMillan et al., 2011a). The quantification of model structural uncertainty is more challenging, since it takes a considerable amount of time and effort to set-up and run several model structures. Furthermore, it is difficult to link intermodel differences to alterations in certain hydrological process formulations (Clark et al., 2008), because models often differ in several process formulations. This is where the use of modular modelling frameworks (MMF), a tool which facilitates switching between model structures (Addor and Melsen, 2019), might provide ways forward in the evaluation of model structural uncertainty. In a MMF it is possible to alter a minor part of the model structure, which allows the researcher to isolate choices in the model development process (Knoben et al., 2019). The Framework of Understanding Structural Errors (FUSE, Clark et al.,

2008) is an example of a modular modelling framework, which can be used to diagnose differences in hydrological model
structures.

This study is designed to evaluate the impact of hydrological model structure on the magnitude and timing of simulated extreme runoff events with different return periods. We combine two state-of-the-art approaches: the hydrological modular modelling framework FUSE, and large ensemble meteorological simulations. The forcing data-set consists of 2,000 years of daily meteorological data, representing the present-day climate conditions. This data set will be used to force several hydrological models within the FUSE framework. The different model structures will be used to evaluate various hydrological process formulations, to determine which process formulations have the largest impact on the simulated magnitude and timing of extreme high- and low-flow events in different climate zones. Due to the length of the forcing time series, the extreme runoff events in the tail of the distribution can be evaluated using simulated values. Hence, we do not rely on statistical models to extrapolate extreme events. As such, this study contributes to the understanding of the impact of model structural uncertainty in hydrological models on simulated extreme runoff events.

## 2 Methods

We assessed the impact of hydrological model structural uncertainty on extreme runoff events by using large ensemble meteorological simulations in combination with the hydrological modular modelling framework FUSE. We examined four different climate zones, because hydrological processes vary considerably between climates (Pilgrim, 1983), which leads to different processes being of importance in controlling the extreme events (Di Baldassarre et al., 2017; Eagleson, 1986). In the R-version of FUSE (Vitolo et al., 2015), ten different model structures were employed, and to represent the complete parameter space, 100 parameter sets were used for every model structure. The simulated extreme runoff events were compared based on their magnitude and timing.

### 2.1 Meteorological forcing data

We employed a 2,000 year time series of meteorological data, generated by the EC-Earth global coupled climate model (v2.3, Hazeleger et al., 2012). This 2,000 year time series originally consisted of a large ensemble of 400 sets of 5 year runs. In this study, these 400 sets were assumed to be one long time series, which enables extensive return period analysis. This time series represents a period with a simulated absolute Global Mean Surface Temperature (GMST) equal to the observed GMST in the years 2011-2015 based on HadCRUT4 data (Morice et al., 2012). The time series thus represents present-day climatic conditions. In Van der Wiel et al. (2019), this data-set was used to evaluate the benefits of the large ensemble technique for hydrology. Further details on the design of the meteorological forcing data are provided in that paper.

The 2,000 year meteorological time series as employed by Van der Wiel et al. (2019) has global coverage at 1.1 $^\circ$ horizontal resolution. However, for this study we restricted ourselves to four climate zones, represented by one grid cell for each climate

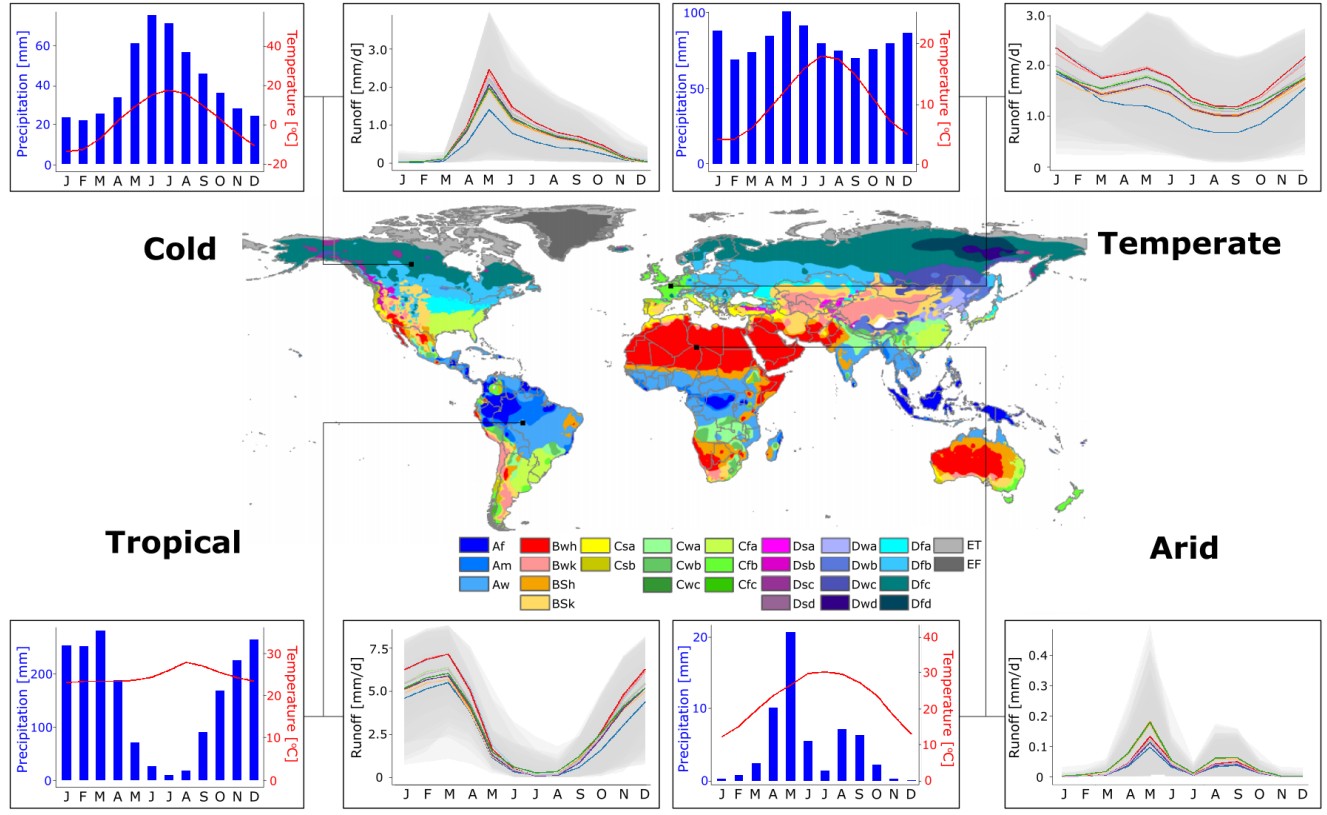

**Figure 1.** Köppen-Geiger climate map indicating the locations of the selected grid cells for the four different climate zones, and their corresponding climatology and hydrology (central map taken from Peel et al. (2007), their Figure 10). The climate graphs show simulated climatological monthly precipitation sums (blue bars) and monthly average temperatures (red lines). The hydrological conditions are visualised using simulated monthly average runoff levels. The different line colours represent the ten evaluated model structures and the spread induced by the different parameter sets is shown using grey bands.

zone (Figure 1). Simulated monthly averaged 2 m temperatures and precipitation sums were obtained from the EC-Earth model to classify grid cells based on the Köppen-Geiger criteria, and allow the selection of appropriate grid cells for this study. We selected four grid cells to represent the arid (BWh), cold (Dfc), temperate (Cfb) and tropical (Aw) climate. This set of climate zones offers a comprehensive representation of the global climate zones (Kottek et al., 2006; Peel et al., 2007).

95

Daily 2 m temperature, precipitation and potential evapotranspiration data for the full 2,000 years were then acquired for the four selected grid cells. The 2 m temperature and daily precipitation fluxes were directly available from the EC-Earth model. Potential evapotranspiration fluxes were calculated following the Penman-Monteith method (Zotarelli et al., 2015). The precipitation and potential evapotranspiration fluxes were used as input in the FUSE models, the 2 m temperature was used to

100 force the snow module (see Section 2.2).

## 2.2 Framework of Understanding Structural Errors (FUSE)

FUSE is a modular modelling framework, which can be used to diagnose differences in hydrological model structures (Clark et al., 2008). FUSE is developed based on four parent models; the U.S. Geological Survey's Precipitation-Runoff Modelling System (PRMS, Leavesley, 1984), the NWS Sacramento model (Burnash et al., 1973), TOPMODEL (Beven and Freer, 2001) and different versions of the Variable Infiltration Capacity (ARNO/VIC) model (Liang et al., 1994). This framework enables the assessment of intermodel differences in another way compared to other model intercomparison studies (Henderson-Sellers et al., 1993; Reed et al., 2004). In FUSE, each model component can be adapted in isolation and therefore the effect of specific hydrological process formulations can be investigated. In the next subsection we further discuss which model structures we selected and which process formulations were tested.

All model structures used in this study were lumped hydrological models, which were run at a daily time step. We employed a spin-up period of five years, before forcing the hydrological models with the 2,000 year meteorological time series. The simulated monthly average runoff varied among the evaluated model structures and parameter sets (Figure 1). Therefore, it is essential to select an adequate hydrological model for the simulation of runoff levels, and it will likely be of larger importance when simulating extreme runoff events.

FUSE as implemented in R (Vitolo et al., 2015) does not include a snow module. However, snow storage and snow melt might be important components in the hydrological cycle of the colder climate zones. Therefore, a snow module was implemented. First, a threshold temperature was defined at 0 °C, below which precipitation is assumed to fall as snow. Secondly, snow melt is simulated by using a simple degree-day method (Kustas et al., 1994):

$$M = a(T_a - T_b), \tag{1}$$

in which $M$ represents snow melt (cm/day), $a$ the degree-day factor (cm/°C/day), $T_a$ the average daily temperature (°C) and $T_b$ the base temperature (°C). The range in the degree-day factor is between 0.35 and 0.60 under most circumstances (Kustas et al., 1994). Therefore, the degree-day factor was fixed at a value of 0.475 cm/°C/day, and $T_b$ was set to 0°C. The degree-day method employed daily 2 m temperature data to subdivide the precipitation data into rain and snow and to determine the melt rate. The different FUSE model structures were subsequently forced by these subdivided precipitation fluxes. The degree-day parameters were kept constant across the experiments, because we only explore one snow formulation (in contrast to the other processes, for which model formulations were all varied).

### 2.2.1 Selected model structures

In total, 1248 different model structures can be constructed in FUSE as implemented in R (Vitolo et al., 2015) by combining different hydrological process formulations from the parent models. The architecture of the upper and lower layer can be altered, and the process formulations for simulating base flow, evaporation, percolation, surface runoff, interflow and routing can

be changed. The lower layer architecture is intimately tied to the process formulation of base flow. Therefore, they need to be changed simultaneously and only a few combinations are possible.

**Table 1.** The model structures that were employed in this study. Each letter refers to a specific hydrological process formulation as in Clark et al. (2008), the model IDs are described by Vitolo et al. (2015). The model abbreviations are related to the alteration in the model structure and are used throughout this paper.

| Model Component | Model Number | | | | | | | | | |
|---|---|---|---|---|---|---|---|---|---|---|
| | 1 | 2 | 3 | 4 | 5 | 6 | 7 | 8 | 9 | 10 |
| **Upper Layer** | A | B | C | C | C | C | A | A | A | A |
| **Lower Layer** | A | A | A | C | B | B | B | B | C | C |
| **Base Flow** | A | A | A | B | C | C | C | C | B | B |
| **Evaporation** | A | A | B | B | A | B | A | A | A | A |
| **Percolation** | C | C | C | C | C | C | C | B | B | B |
| **Interflow** | A | A | A | A | A | A | A | A | A | A |
| **Surface Runoff** | A | A | A | A | B | B | A | A | A | B |
| **Routing** | A | A | A | A | A | A | A | A | A | A |
| **Model ID** | 802 | 800 | 642 | 626 | 808 | 652 | 790 | 880 | 874 | 896 |
| **Abbreviation** | UL1 | UL2 | LL1 | LL2 | EV1 | EV2 | PC1 | PC2 | SR1 | SR2 |
| **Alteration** | Upper Layer | | Lower Layer | | Evaporation | | Percolation | | Surface Runoff | |

Ten different model structures were evaluated in this study. Table 1 provides an overview of the selected hydrological model structures. In the odd model numbers, new model structures were constructed and in the even model numbers, a single hydrological process was altered in the model structure relative to the preceding odd model number. By comparing the extreme runoff events simulated between consecutive odd and even numbered model structures, we analysed the impact of a specific hydrological process on extreme event simulation, indicated by the alteration in Table 1.

For our synthetic experiment, we decided to apply a fixed routing scheme. The effect of routing parameters on the discharge signal is delay and attenuation. As such, the main effect of the routing scheme would be to decrease the peak height. Since we evaluate our model results on (amongst others) peak height, the routing would dominate the results without providing insights on the underlying runoff-generating processes. Besides routing, the process formulation of interflow was left unchanged throughout this study, as it was not explicitly parameterised in TOPMODEL and ARNO/VIC (Clark et al., 2008).

In contrast to other studies that evaluate different model structures (Atkinson et al., 2002), this study evaluated differences among model structures that are deemed to be equally plausible. Hence, there were no prior expectations of specific models to outperform other models. This means that the emphasis in FUSE is not on the lacking parts of hydrological models, but on the intermodel differences that are caused by different representations of the real world (Clark et al., 2008).

### 2.2.2 Parameters

In this synthetic experiment, the parameters of the hydrological models were not calibrated to real catchment observations. Instead, the parameters of the models were sampled over their full range. Since in calibrated experiments it is always difficult to differentiate the effect of parameter values from the effect of model structure, the parameter sampling approach also created the opportunity to assign the effect on extreme events either to parameter values or to model structure.

To investigate the appropriate and feasible number of parameter sets required to sufficiently capture parameter space, the Kolmogorov-Smirnov test was employed (Massey Jr, 1951). With the Kolmogorov-Smirnov test, we compare the difference in the distribution of the hydrological model output between a small parameter sample and a large benchmark sample. Our benchmark sample had a size of 5000 parameter sets. We applied the Kolmogorov-Smirnov test to assess the annual maximum and minimum daily runoff from 10 up to 200 parameter sets, each time with 10 samples increment. The model runs were executed for 30 years to save computation time, because this is considered sufficient to represent the mean climate conditions (McMichael et al., 2004). The D-statistic describes the largest distance between the Empirical Cumulative Distribution Functions (ECDF), which indicates that when the D-statistic decreases, the ECDFs are more likely to originate from the same data-set.

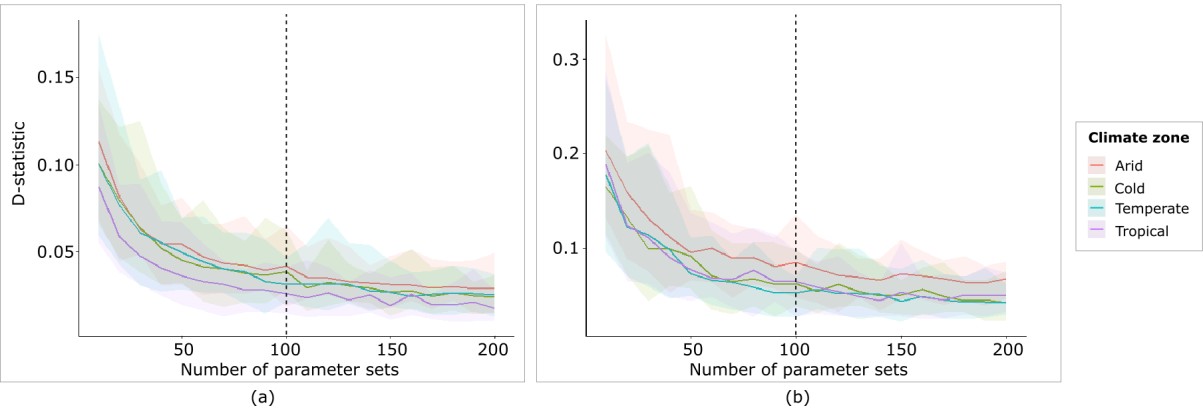

**Figure 2.** D-statistics for maximum (a) and minimum runoff (b) in one model structure (UL1) with twelve parameters, which result from the Kolmogorov-Smirnov test. The other model structures show a similar trend (not shown). The bands are a result of the different parameter samples, the different colours represent the four climate zones.

We found that the optimal trade-off between computer time and sufficiently capturing parameter space was at 100 parameter sets, as the D-statistic stabilised at this value (Figure 2). Since there are different process formulations, the number of sampled parameters varied between eleven and fifteen for the different model structures. Nevertheless, for justification we used 100 parameter sets for all model structures, independent of the number of parameters. The parameter sets were generated using Latin Hypercube Sampling, based on the parameter ranges provided by Vitolo et al. (2015), as given in Table A1.

### 2.3 Magnitude of extreme runoff events

The magnitudes of the simulated extreme events were evaluated by comparing the distribution of runoff values based on four return periods: 25, 50, 100 and 500 years. The associated runoff levels were determined by sorting the time series of annual maximum and minimum daily runoff values. This resulted in 2,000 sorted runoff values from which events were selected. For instance, for the 500-year return period, the 4th most extreme value in the sorted time series was taken.

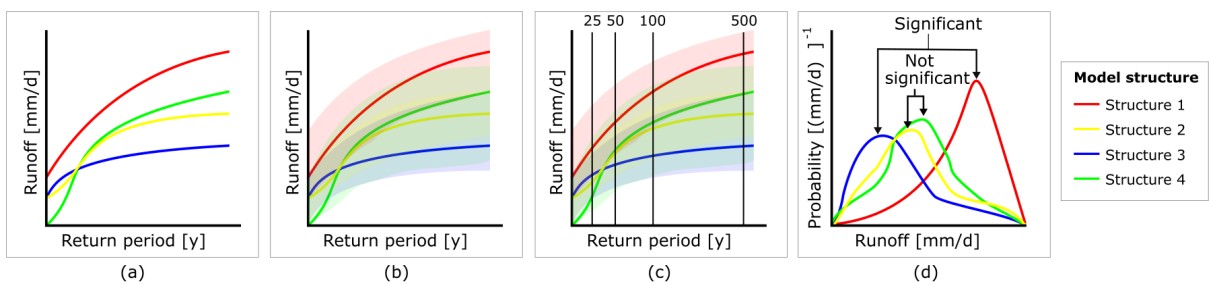

**Figure 3.** Illustration of the conducted procedure for the comparison of the extreme event magnitude. The four lines with different colors represent the different model structures (theoretical structure 1-4). (a) The simulated runoff can be plotted against return period for the different models. (b) The uncertainty bounds are due to the 100 different parameter samples per model. (c) The parameter samples were compared at different return periods. (d) The projected difference between the distributions at a given return period of the various model structures was tested using a two-sample t-test, an example of a significant and a not-significant difference is shown.

The different model structures yielded different simulated magnitudes for extreme runoff events (Figure 3a). Every model structure was run using 100 different parameter sets, which led to bands around the projected extreme runoff events (Figure 3b). The runoff values and their bands were subsequently evaluated for 25, 50, 100, and 500 year return periods (Figure 3c). The different parameter sets resulted in 100 extreme runoff values at a specific return period for every model structure. In order to test whether the projected difference in the distributions of these runoff values (Figure 3d) was significantly different from the paired model, a two-sample t-test was applied. This test was used to evaluate related model structures based on a change in one single hydrological process formulation (Table 1). By comparing related model structures, the impact of corresponding hydrological process formulations could be isolated for specific climate zones and return periods.

For the magnitude analysis of low-flow events, we encountered that some combinations of model structures and parameter sets led to a very low fixed value (in the order of $10^{-4}$ and less), which we refer to as hard-coded lower limits. These lower

limits varied between model structures, dependent on the configuration of different storage reservoirs. These limits assure
numerical stability, but could obfuscate our analysis, because the difference between distributions simulating lower limits would be significant if the lower limits between two model structures had different values. Conceptually, the lower limits represent zero discharge: the river has run dry. As such, no significant different should be found when two models reached this lower limit. Therefore, in all simulations the lower limit in discharge was set equal to zero.

## 2.4 Timing of extreme runoff events

An asset from the ensemble approach for return period evaluation compared to GEV statistics, is that it also allows us to evaluate the timing of the 500-year events based on the entire 2,000 year time series. Extreme hydrological events do not always result from extreme meteorological conditions, but could also originate from a sequence of moderate weather conditions (Van der Wiel et al., 2020). By assessing the timing of extreme runoff events, we investigated whether the timing of the extreme runoff events is controlled by different model structures and parameter sets or determined by the meteorological forcing.

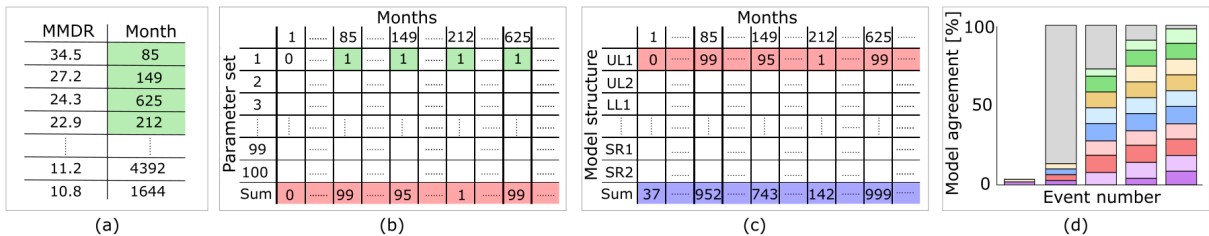

(a) (b) (c) (d)

**Figure 4.** Illustration of the procedure for the comparison of the timing of extreme events, equal or greater than 500-year events. (a) In each simulation, the monthly maximum daily runoff (MMDR) values were sorted and the four most extreme events were selected (green cells), this table shows an example for one parameter set. (b) All parameter sets of one model structure were concatenated and the sum (red cells) indicated the variation in timing in one model structure. A score of 100 means that for all different parameter sets, the same event is selected. (c) All model structures of one climate zone were concatenated and the sum (blue cells) indicated the variation in extreme event timing in all model structures for one climate zone. The values in the blue cells have a maximum score of 1000 (10 models, with 100 parameter samples each). A score of 1000 indicates that all models and all parameter sets identify the same event as a 500-year event (100 % model agreement). Given that we have 2,000 years of simulations and evaluate 500-year events, the ideal case where all models agree would result in four events with a score of 1000. (d) Stacked bar charts are used to visualise the model agreement of specific runoff events. Each row in (c) represents one colour in the coloured bars in (d), the total height of each bar is determined by the value in the blue cells in (c). Grey shading behind the bars indicates the theoretical maximum for 500-year events: four runoff events with 100 % model agreement.

The timing of extreme runoff events with 500-year return periods were compared. This was done in four steps, as depicted in Figure 4.

First, we sorted all the monthly maxima and minima daily runoff values and their corresponding simulation month (Figure
4a). The four most extreme events in this sorted 2,000-year data-set represent the extreme events equal or greater than the 500-
year events. These four most extreme events were determined for each model simulation, so for each combination of model
structure and parameter sets.

Then, we evaluated to what extent the same events were selected for different parameter sets, but with the same model
structure. If one event was for instance selected for all 100 parameter sets, this particular event would have a score of 100 in
the red row of Figure 4b. If this event was only selected for half of the parameter sets, it would have a score of 50. If across
all parameter sets the same four events would be identified, this would result in four times a score of 100 in Figure 4b. This
indicates that the influence of hydrological parameters on the timing of the extreme event is negligible.

This procedure was repeated across all 10 model structures. If the same event was selected for all parameter sets (n=100)
and for all models (n=10), it would result in a score of 1000 in the blue row of Figure 4c. If the same four events were selected
across all models and all parameter sets, four times a score of 1000 would be found. In that case, both model structure and
model parameters have negligible influence of the timing of the extreme event: the event is mainly triggered by meteorological
circumstances.

Finally, the model agreement of the specific extreme runoff events was evaluated in stacked bar charts (Figure 4d). The
colours of the stacked bars represent the different model structures and the height of these bars indicates the model agreement
within a specific model structure for different parameter values. For instance, in Figure 4d, one event is identified by almost all
simulations and it approaches a fully coloured bar chart. The percentage of model agreement was determined by the amount
of model simulations that identify a specific extreme runoff event out of a total of 1000 model simulations, where all model
simulations employed a unique combination of a model structure and a corresponding parameter set.

If all combinations of different model structures and parameter sets agree upon the timing of this extreme event, only four
events would be identified in total. This would lead to the theoretical maximum, where there are four fully filled stacked bar
charts and an x-axis going to a maximum of four. This would be shown by four fully stacked bar charts, in other graphs shown
by grey shading. When the simulation do not agree upon the timing, there will be more bars in the chart, indicating the variation
in the timing. The value of the x-axis indicates the total number of selected extreme events. For example, a value of 20 on the
x-axis indicates that across all simulations, 20 different 500-year events with a different timing were identified. We sort the
selected events by model agreement, with events of bigger model agreement (higher bars) shown on the right.

# 3    Results

## 3.1    Magnitude of extreme runoff events

This section describes the impact of model structures on extreme event magnitude for different climate zones, hydrological process formulations and return periods. We compared the distribution of the magnitudes of the extreme high- and low-flow events for related model structures, based on four different return periods, and for four different climate zones. Alterations in the hydrological process formulations lead to a difference in the magnitude of extreme runoff events, as depicted in Figure 3a, which is an example showing the principle. Figure 5 shows the same information, but based on actual simulations of high-flows (Figure 5a&c) and low-flows (Figure 5b&d) in the tropical climate zone. We then employed a two-sample t-test to calculate the p-values (Figure 6), which were used to distinguish the statistically significant (p ≤ 0.05) and non-significant (p > 0.05) differences in the distribution of extreme event magnitudes as in Figure 3d.

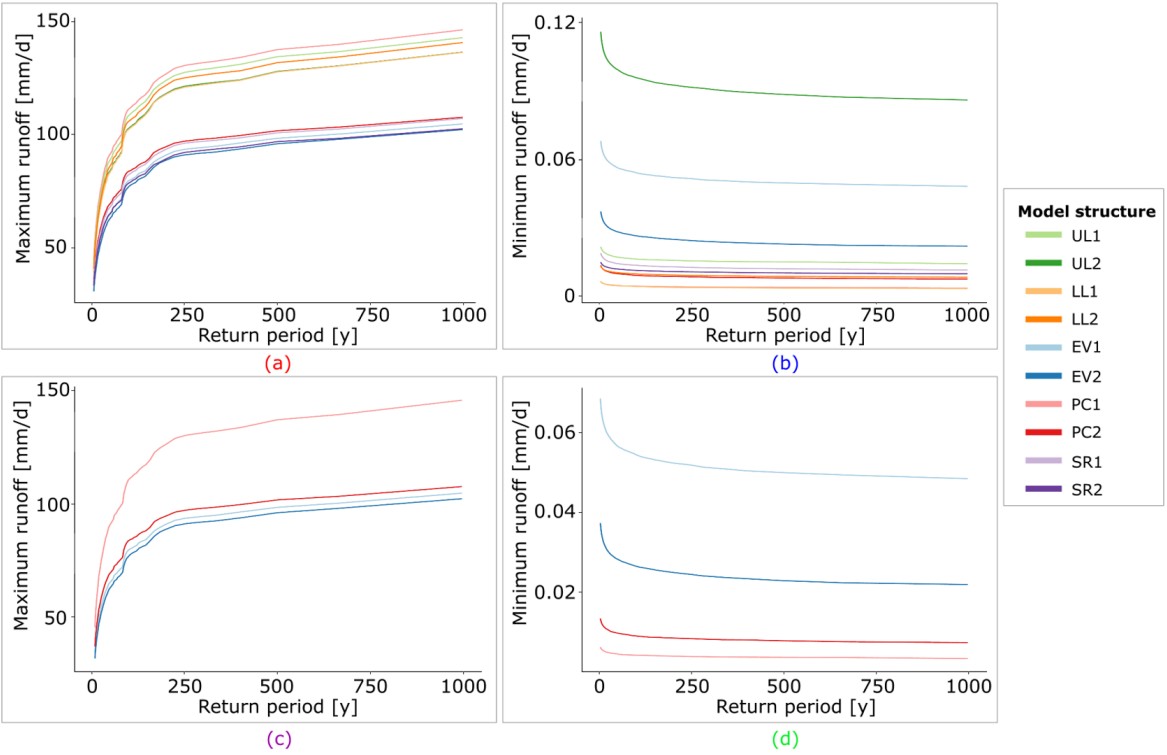

**Figure 5.** The ensemble mean of the annual maximum (a&c) and minimum (b&d) daily runoff levels at different return periods in the tropical climate zone. The ensemble mean is obtained based on 100 parameter sets. In (a) and (b) all model structures are visualised. In (c) and (d), a selection of only four model structures is presented to emphasize the difference between the model structures. These four model structures are related by alterations in the evaporation (EV1, EV2) and percolation (PC1, PC2) process formulations (Table 1). The colours of the panel labels refer to the boxes of the same colours in Figure 6.

Panel c and d of Figure 5 highlight four models in the tropical climate zone for comparison. The model structures that are related by an alteration in the process formulation of percolation, simulate a difference in extreme high-flow magnitude for all return periods (red lines in Figure 5c). Based on the t-test conducted on these distributions, this results in a significant impact of alterations in the process formulation of percolation for all return periods (as displayed in Figure 6). In contrast, the

model structures related by an alteration in the process formulation of evaporation, simulate comparable runoff values across all return periods for high-flows (blue lines in Figure 5c). Therefore, there is no significant impact on the magnitude of extreme high-flow events caused by this hydrological process formulation (Figure 6). For the low-flows, an alteration in the percolation formulation (Figure 5d) does not lead to statistically significant differences in the low-flow distribution (Figure 6), whereas an alteration in the evaporation formulation leads to a difference at the 0.1-significance level.

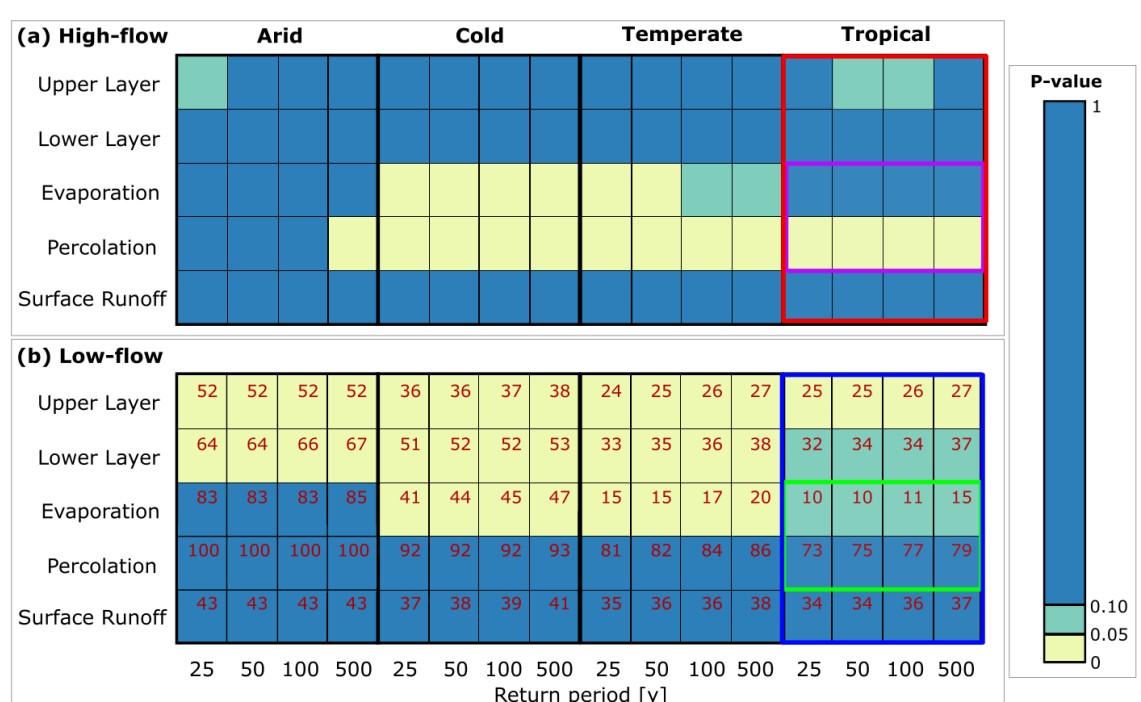

**Figure 6.** Statistically significant ($p \leq 0.05$) and non-significant ($p > 0.05$) differences between the distribution of magnitudes for extreme runoff events, assessed by a two-sample t-test. The colours indicate whether an alteration in the model structure has a statistically significant impact on the magnitude of extreme high- (a) and low-flow (b) events. This is shown for the four climate zones (arid, cold, temperate and tropical, indicated at the top) and the four different return periods (25, 50, 100 and 500 years). The red values in (b) indicate the percentage of simulations which reached zero runoff conditions (dry river). The coloured boxes refer to the results displayed in the panels of Figure 5.

An alteration in the model structure has significant impact in about a quarter of the model output comparisons during high-flow events (Figure 6a). The difference between the magnitude distributions of the high-flow events is non-significant for

alterations in the architecture of the upper and lower layer and in the process formulation of surface runoff. This means that the magnitude of high-flow events for all climate zones and return periods is not sensitive to changes in the formulation of these hydrological processes.

In the arid climate zone, the impact of alterations in model structures on high-flow events has the least impact. This indicates that the magnitudes of the high-flow events are mainly controlled by the meteorological forcing. In the cold and temperate climate zones, the high-flow events are sensitive to alterations in the process formulation of two hydrological processes; evaporation and percolation. This indicates that the magnitudes of the high-flow events are not only determined by the meteorological forcing, but there is also a notable impact of the hydrological model structure, specifically for the formulation of these two processes. Finally, in the tropical climate zone, the high-flow events are only sensitive to alterations in the process formulation of percolation. The other hydrological process formulations do not significantly affect the magnitude of high-flow events in this climate zone.

For low-flow events, the model structure has a greater impact on the simulation of extreme events. An alteration in the model structure has significant impact in half of the model output comparisons during low-flow events (Figure 6b). In the arid climate zone, the low-flow events are sensitive to alterations in the architecture of the upper and lower layer. In the cold and temperate climate zones, the low-flow events are also sensitive to alterations in the architecture of the upper and lower layer, and additionally to changes in the process formulation of evaporation. In the tropical climate zone, the low-flow events are less sensitive to alterations in the architecture of the lower layer and the process formulation of evaporation, but still sensitive to alterations in the upper layer architecture. In most climate zones, changing the formulation of multiple hydrological processes significantly impacts the simulation of the magnitude of low-flow events, which implies that the model structure is an important source of uncertainty. The meteorological forcing is clearly not the only factor controlling the magnitude of simulated low-flow events.

A phenomenon that does play an important role in the evaluation of low-flows is that eventually in some cases, the simulated runoff goes to zero, indicating that no more water is flowing through the river. For instance in the arid climate zone, for the two models where percolation is altered, 100% of the simulations have zero discharge already for the 25-year return period events. Differences in low-flows as a consequence of changing the percolation formulation can then no longer be traced and thus do not lead to a significant difference.

## 3.2 Timing of extreme runoff events

The timing of extreme high-flow events is evaluated using stacked bar charts. Figure 7 shows the percentage of model agreement on the timing of extreme high-flow events with a return period equal or greater than 500-years, as earlier depicted in Figure 4d. For the low-flow events, the timing evaluation could not be conducted, because of the nature of low-flow events to

persist longer. This will be further discussed in this section.

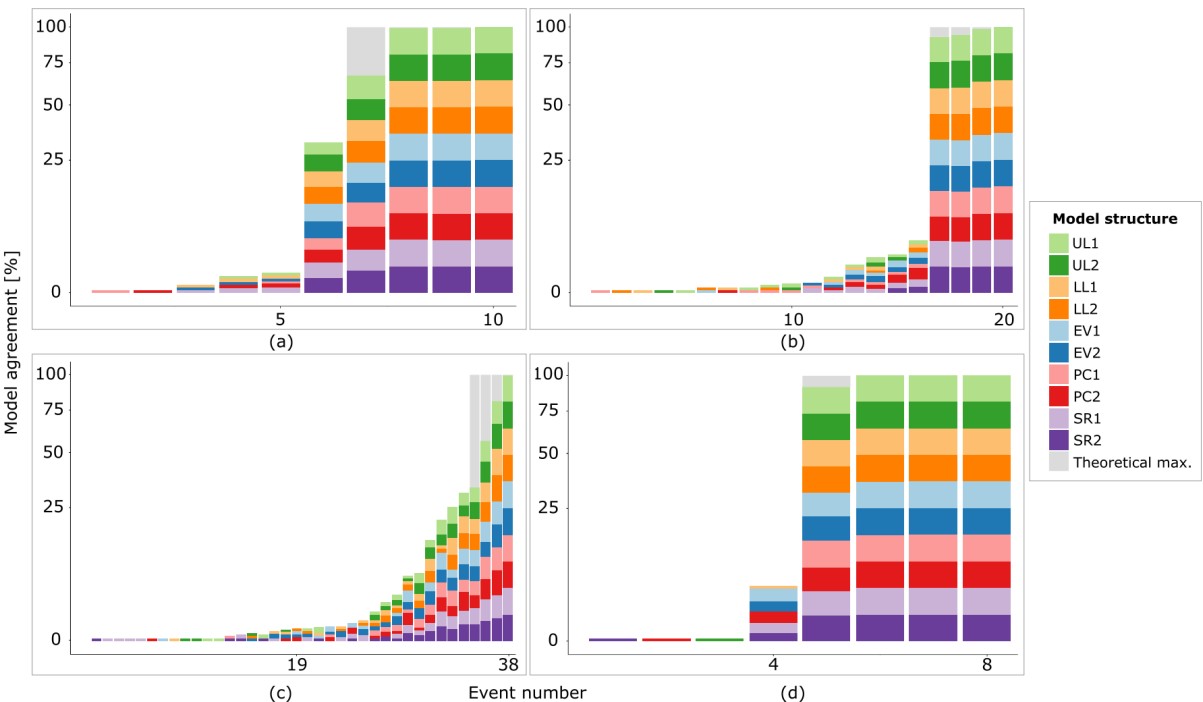

**Figure 7.** Stacked bar charts that visualise the percentage of model agreement for extreme high-flow events. Four different climate zones are evaluated in the different subplots; arid (a), cold (b), temperate (c) and tropical (d). Extreme runoff events are identified when they are equal or greater than the 500-year return level. The linked model structures (Table 1) are related to each other by comparable colours. Grey shading indicates the theoretical maximum of four events with 100 % model agreement, which would imply a negligible impact of model structure and parameters on high-flow event timing.

The impact of different hydrological process formulations and parameter sets on the timing of extreme high-flow events varies between the selected climate zones. In the arid and tropical climate zones, there are multiple events with a model agreement exceeding 99 %. In these cases, almost all model simulations agree on the timing of these extreme events. Just ten and eight runoff events were selected (out of a total of 24,000 potential events) as extreme high-flow events in the arid and tropical climate zones, respectively (Figure 7). This means that there are only a few model simulations that deviate by simulating the most extreme runoff events at a different point in the time series. For these climate zones, this implies that the timing is mainly prescribed by the meteorological forcing. This might be explained by the precipitation climatology in these climate zones. On average, in the arid climate zone the daily precipitation sum exceeds 1 mm only during eleven days a year. Precipitation is therefore scarce and characterised by short events of high-intensity (Goodrich et al., 1995), which propagate into extreme runoff events. In the tropical climate zone, there is a high precipitation rate throughout the complete time series. However, there is a pronounced wet season from October until April (Figure 1). There are multiple extreme precipitation events larger than

150 mm/d. The 500-year extreme runoff events are initiated by these extreme precipitation events.

In both the cold and temperate climate zone, there is only one event with a model agreement exceeding 99 % (Figure 7). In the cold and temperate climate zones, there are 20 and 38 different events selected as extreme events, respectively. The selected runoff events with the highest model agreement are initiated by the most extreme precipitation events, whereas the selected extreme runoff events with a low model agreement are most likely initiated by compound events (Van der Wiel et al., 2020; Zscheischler et al., 2018). Hence, the timing of extreme high-flow events may depend more on hydrological processes, and consequently vary across hydrological model structures and parameter values in these climate zones. The stacked bar charts indicate which model structures lead to the selection of events with low agreement. Some model structures seem to show deviant behaviour, but there is no convincing pattern visible; most model structures seem to be represented in low-agreement events. Therefore, there is no clear relationship between the extreme runoff events with a low model agreement and specific model structures. We hypothesise that this uncertainty can be assigned to the difference in parameter sets.

To evaluate the timing of extreme low-flow events, a similar approach was applied compared to the high-flow events. However, for several combinations of model structures and parameter sets, zero runoff was simulated (Figure 5b). These periods of zero runoff often persisted for longer time periods, and therefore, it was not possible to select the four most extreme events. This invalidates our method to investigate the impact of different model structures on the timing of low-flow events, at least with the definition of low-flow events as we employ it (directly evaluating the runoff). Zero runoff, representing a dry river, is mostly occurring in the drier climate zones. In the arid climate zone, the runoff levels drop to zero in 69 % of all the model simulations. In the cold climate zone, 53 % of the model combinations simulate zero runoff. In this climate zone, the temperature regularly drops below zero degrees C (Figure 1), which indicates that precipitation falls as snow instead of rain. This transition affects the runoff-generating processes (Immerzeel et al., 2009), which results in lower runoff levels during colder periods (Figure 1). In the temperate and tropical climate zones, 39 % and 36 %, respectively, of all combinations of model structures and parameter sets simulate zero runoff.

## 4  Discussion

This study evaluates the spread introduced by different hydrological model structures and parameters on the magnitude and timing of simulated extreme runoff events. Both differences and similarities can be identified between the distributions of runoff values for the high- and low-flow events. Alterations in hydrological model structures more often result in significant differences in low-flows (45 %) compared to high-flows (24 %), which implies a larger model structural uncertainty in the magnitude of low-flow events. High-flow events mainly depend on precipitation, i.e. meteorological forcing, while the influence of other runoff generating processes such as soil moisture and base flow is marginal (Zhang et al., 2011). This is not to say that these processes are not relevant: merely, our results demonstrate that the way these processes are formulated in the model has limited impact on the model result. The situation during high-flow events is often characterised by a precipitation surplus. Therefore,

there will be more or less continuous groundwater recharge by percolation in the unsaturated zone (Knutsson, 1988), which explains why the formulation of percolation appears as a relevant hydrological process to estimate the magnitude of high-flow events.

Hydrological models are traditionally designed to simulate the runoff response to rainfall and therefore, it seems to be more challenging to simulate low-flow events (Staudinger et al., 2011). The low-flow events are mainly sensitive to alterations in the architecture of the upper and lower layer. Earlier research indicates the importance of the lower layer architecture and the process formulation of base flow in simulating low-flow events (Staudinger et al., 2011). The architecture of the upper and

345 lower layer defines the water content in these layers (Clark et al., 2008). This water content is controlling the runoff-generating processes during low-flow events due to a precipitation deficit and reduces the importance of the percolation process (Andersen et al., 1992). Therefore, alterations in the process formulation of percolation mainly affect high-flow events in the wet climate zones. Alterations in model structures have no distinct impact on the simulated timing of high-flow events. Nevertheless, there is a spread in the timing of these events, which is most likely caused by the difference in parameter sets.

Besides these differences, there are also similarities in the simulation of high- and low-flow events. The magnitudes of high- and low-flow events in the cold and temperate climate zone show significant differences for the same alterations in the hydrological process formulations. Furthermore, alterations in the process formulation of surface runoff have no significant impact on the magnitude of both types of extreme runoff events. This might be due to the lacking implementation of infil-

355 tration excess overland flow in FUSE (Clark et al., 2008). This could be an important factor for surface runoff, especially in arid climate zones (Reaney et al., 2014). Another factor might be the temporal resolution of the model runs: the models are run at a daily time step, while surface runoff is especially relevant at shorter time steps (Morin et al., 2001; Melsen et al., 2016).

In the next sections we synthesize the results per climate, discuss our study design, and make a note about translating hy-

360 drology to societal impacts.

## 4.1 Climate synthesis

The magnitude and timing of the extreme high-flow events in the arid climate zone are mainly controlled by the meteorological forcing. This is contrary to previous studies in which the runoff in dry catchments was more sensitive to different hydrological

models (Jones et al., 2006; Lidén and Harlin, 2000), but here we specifically refer to high-flow events in arid climates. In this climate zone, precipitation is scarce and often characterised by extremely variable, high-intensity and short-duration events (Goodrich et al., 1995). Consequently, runoff in arid climate zones is characterised by a dominance of Hortonian overland flow (Segond et al., 2007). This runoff-generating process is not included in the implementation of FUSE, which might reduce the impact of different model structures (Clark et al., 2008). Also the temporal resolution at which we ran the model and evaluate

the high-flow events might be relevant. The extremely flashy precipitation patterns can cause flash floods that occur over the

course of a few hours. We evaluate the model results at the daily time step, which can cover up the occurrence of flash flood events. For the low-flow events, we found more spread in the magnitude as a consequence of altering process formulations. Alterations in multiple hydrological processes result in significant differences.

In the cold and temperate climate zones, there is more spread in the simulations regarding the magnitude and timing of extreme runoff events. The magnitudes of extreme high- and low-flow events are sensitive to alterations in multiple hydrological process formulations, which implies that several hydrological processes are important in the runoff-generating processes in these climate zones, as also discussed by Scherrer and Naef (2003). In different model simulations different high-flow events are identified as most extreme runoff events, which leads to a spread in the timing of these events. This spread is partly assigned

to the difference in parameter sets.

We only tested a limited amount of processes and process formulations. However, especially in the cold and temperature climate zones, extreme events related to snow melt can potentially occur. Therefore, the process formulation of snow melt could have significant impact on the simulations. This was, however, not tested because we only used a single degree-day snow

formulation. The results are therefore conditional on the processes that we altered, and that were available within the FUSE framework.

In the tropical climate zone, the spread in the magnitude and timing of extreme runoff events is small, which indicates that the extreme events are mainly controlled by the meteorological forcing. There is only one process formulation that simulates

a significant impact on the magnitude; percolation for the high-flows and the upper layer architecture for the low-flow events. The formulation of the percolation process controls the high-flow events in the tropical climate zone, as there are months with large amounts of precipitation (Figure 1). Due to these large amounts of precipitation, water is subjected to percolation through the succeeding layer (Bethune et al., 2008; Savabi and Williams, 1989). The role of the upper layer architecture in the simulation of low-flow events might be related to evaporation dynamics - although the evaporation formulation has less significant

impact ($0.05 < p \leq 0.1$).

We found no distinct relationship between the length of return periods and the degree of uncertainty in the magnitude of extreme runoff events. There are situations in which the difference between related distributions of high-flow events become significant when the length of the return period increases, e.g. the percolation process formulation in the arid climate zone.

There are also distributions of related model structures that are significantly different at shorter return periods, e.g. the evaporation process formulation in the temperate climate zone. This contrast might be explained by the difference in importance of specific hydrological processes or parameters for events at different return periods.

## 4.2 Study design

We designed a synthetic experiment to conduct controlled experiments on the role of model structure on the simulation of extreme runoff events. There are, however, a few implications when using a synthetic approach. In this study, the models were not calibrated in order to isolate the impact of different model structures. It is however common practice to use a pre-defined model structure, which is fitted to the local circumstances via parameter calibration (McMillan et al., 2011b). In this study the complete parameter range was sampled: all combinations of parameter values were considered equally plausible and interde-

pendence of parameters was not considered since we used the Latin Hypercube Sampling approach (Clarke, 1973; Helton and Davis, 2003). Tuning the parameters to a specific location could reduce the parameter range, and smaller parameter ranges could lead to more realistic runoff values (Cooper et al., 2007), which might have revealed a relatively higher impact of model process formulation on model results. This, however, comes at a loss of generality. Also, when calibrating hydrological models to simulate extreme runoff events, other challenges remain. Especially the limited availability of historical observations can

create a problem for the reliable calibration of extreme events (Wagener et al., 2010); since many observation records do not exceed a length of 50 years, models are forced to simulate outside of their calibration range. This will negatively influences model performance, as for instance demonstrated by Imrie et al. (2000).

The 2,000-year meteorological time series used in this study originally consists of a simulated large ensemble of 400 sets

of 5-year runs. These 400 sets were concatenated artificially. This concatenation might lead to strange transitions of meteorological conditions once every 5 years, as the December month is followed by the next January month of a new 5-year set. Nevertheless, we decided to treat this large ensemble as a single time series, in order to allow for extensive return period analysis. We consider the effect of the concatenation limited since we only evaluate the annual and monthly maximum and minimum daily runoff levels. The employed time series does not allow for the evaluation of multi-year low-flow events, despite these

events being extremely relevant considering their societal impact.

Besides choices in the sampling strategy and choices in the treatment of the meteorological forcing, we also made choices in the characteristics of high and low flow events that we evaluated. Because this is a first extensive exploration of the role of model structure on the simulation of extreme events with long return periods, we evaluate high- and low-flows for their

most straight forward characteristic: the maximum and the minimum runoff. There are, however, ample other characteristics that could be of relevance in the context of hydrological extremes. For high-flow events, besides peak height and timing, also volume is a frequently evaluated characteristic (Lobligeois et al., 2014), while for low-flow events duration and volume deficit are other frequently applied characteristics (Tallaksen et al., 1997). Our approach, being a combination of long-term meteorological simulations and a modular modelling framework, can easily be extended to these characteristics.

Model selection is a crucial step in hydrological modelling. Different hydrological models might lead to substantially different outcomes (Melsen et al., 2018). When hydrologists are familiar with a certain model, they tend to stick to this model,

even though other models might be more adequate for a specific objective (Addor and Melsen, 2019). Model intercomparison studies can provide guidance for model selection and improve model adequacy in the future. This study evaluates the impact of alterations in model structures on extreme runoff events. Some alterations in the model structure lead to significant impacts in the simulation. For example, in the tropical climate zone, the formulation of the percolation process is important. This information can be regarded in model selection of future studies, which will result in more adequate model selection. It should, however, be noted that the framework employed in this study (FUSE) is only representative for a particular suit of bucket-based models. Whereas these models are suitable for long term simulations due to their low data demand and high computational efficiency, results might look different when a more process-based framework, such as SUMMA (Clark et al., 2015a, b), would have been employed.

## 4.3 Societal impact

This study evaluated the translation of meteorology to hydrological extreme impact events. Return periods were used to sort runoff events based on their extremeness, as return periods are frequently used in policy design (Marco, 1994; Read and Vogel, 2015). However, this study does not translate hydrological impact events to the societal impact, which implies that fatalities and economic losses are not examined. This relationship might be affected by non-linear effects, similar to the meteorology-hydrology relationship (Van der Wiel et al., 2020). Therefore, a direction for future research is to link societal impact to return periods of extreme runoff events. The accurate assessment of vulnerability and societal impact requires information related to exposure and sensitivity (Cardona et al., 2012).

## 5 Conclusions

Hydrological extremes are natural hazards that affect a large number of people on a global scale. Several hydrological models were employed to simulate these extremes, with the aim to investigate the impact of hydrological model structure on the simulation of extreme runoff events. The combination of two state-of-the-art approaches, the hydrological modular modelling framework FUSE and large ensemble meteorological simulations to study extreme events, provided insights into uncertainties of the simulations. Parameters of the hydrological models were sampled in a synthetic experiment, which enabled the examination of the impact of different hydrological process formulations on the magnitude and timing of extreme high- and low-flow events, independent of calibration.

The impact of hydrological process formulations on magnitude and timing of extreme runoff events varies among different climate zones (Figure 8). In the arid climate zone, the magnitude and timing of the extreme high-flow events are not affected by changing process formulations or parameter sets. The magnitudes of the low-flow events are significantly affected by alterations in the architecture of the upper and lower layer. In the cold and temperate climate zones, we found a larger spread in the

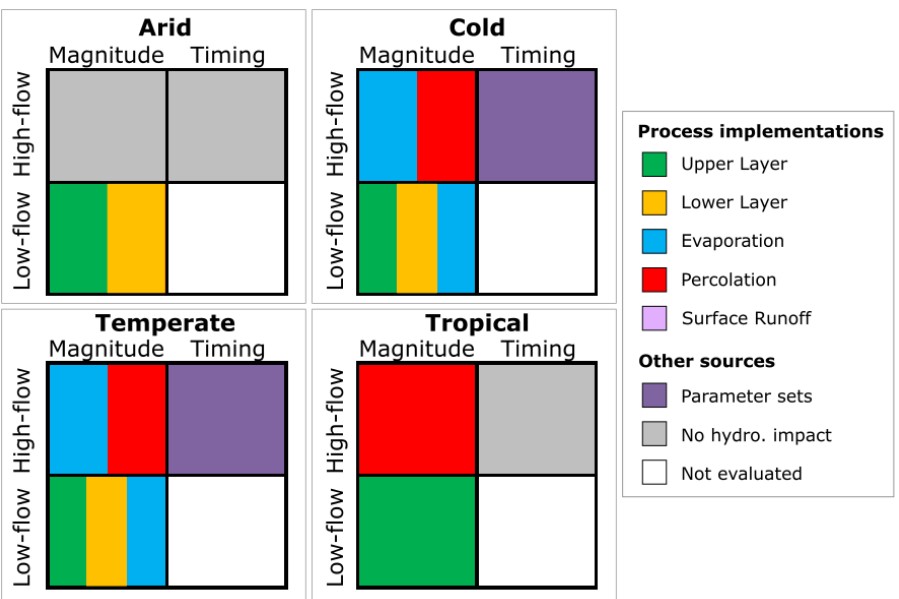

**Figure 8.** Summary of the results: indicated are the process formulations that significantly affect the distributions of the extreme runoff events in the different climate zones. The process implementations refer to the formulation of hydrological processes in different model structures. "No hydro. impact" indicates that the effect of the hydrological model was limited, which implies that neither alterations in the model structure, nor in the parameter sets significantly affected the simulated extreme runoff events.

simulations of the extreme runoff events. Multiple hydrological processes significantly affect the magnitude of the high- and low-flow events, which implies that the model structure is an important source of uncertainty. Therefore, it is essential to select an adequate hydrological model when simulating extreme events in cold and temperate climate zones. Besides that, there is a spread in the timing of high-flow events, caused by different parameter sets in these climate zones. The magnitudes of the high- and low-flow events in the tropical climate zone are affected by the formulation of percolation and upper layer, respectively.

The timing of these events is hardly affected by hydrological model structure or parameter sets, which implies that the timing of these events is dictated by the meteorological forcing. The timing of low-flow events is not evaluated in this study, as many simulations resulted in zero runoff for extended periods.

The results revealed a spread in the simulation of extreme runoff events as a consequence of different hydrological model
structures. The impact of different model structures is larger for the simulation of low-flow events compared to high-flow events. For the low-flow events, hard-coded lower limits were found, implemented for numerical stability. This revealed the numerical challenge that comes with simulating extremely low values. In this study, we interpreted these hard-coded lower limits as zero runoff. The extreme events were assessed at different return periods. However, no clear relationship was found between the model structural uncertainty in the magnitude of extreme runoff events and the return period length.

Insights provided by this study contribute to a better understanding of the importance of the hydrological model formulation of specific processes in different climate zones. These insights can be used in future studies, which will result in more adequate model selection leading to improved understanding and more reliable predictions of extreme runoff events.

*Code and data availability.* All codes to process the data (R-code) and the results themselves are available upon request from the corre-
490 sponding author. The meteorological forcing and all simulated runoff data of the four evaluated climate zones are publicly available; see ... (DOI: XXX).

*Author contributions.* KW and LM designed the study in consultation with GK. KW provided the meteorological forcing data, which GK employed to carry out the hydrological simulations and analyses. GK wrote the manuscript with support from KW and LM.

*Competing interests.* The authors declare that they have no conflict of interest.

*Acknowledgements.* KW acknowledges the HiWAVES3 project, funded by the Netherlands Organisation of Scientific Research (NWO) under Grant Number ALWCL.2 016.2.

**Table A1.** Description and range of the parameters that were sampled, based on Vitolo et al. (2015)

| Parameter | Description | Unit | Values Min | Max |
|---|---|---|---|---|
| $\text{rferr}_{\text{add}}$ | additive rainfall error | mm | 0 | 0 |
| $\text{rferr}_{\text{mlt}}$ | multiplicative rainfall error | - | 1 | 1 |
| frchzne | fraction tension storage in recharge zone | - | 0.05 | 0.95 |
| fracten | fraction total storage in tension storage | - | 0.05 | 0.95 |
| $\text{maxwatr}_1$ | depth of the upper soil layer | mm | 25 | 500 |
| percfrac | fraction of percolation to tension storage | - | 0.05 | 0.95 |
| fprimqb | fraction storage in $1^{st}$ baseflow reservoir | - | 0.05 | 0.95 |
| $\text{qbrate}_{2a}$ | baseflow depletion rate $1^{st}$ reservoir | $\text{day}^{-1}$ | 0.001 | 0.25 |
| $\text{qbrate}_{2b}$ | the baseflow depletion rate $2^{nd}$ reservoir | $\text{day}^{-1}$ | 0.001 | 0.25 |
| $\text{qb}_{\text{prms}}$ | baseflow depletion rate | $\text{day}^{-1}$ | 0.001 | 0.25 |
| $\text{maxwatr}_2$ | depth of the lower soil layer | mm | 50 | 5000 |
| baserte | baseflow rate | $\text{mm day}^{-1}$ | 0.001 | 1000 |
| rtfrac1 | fraction of roots in the upper layer | - | 0.05 | 0.95 |
| percrte | percolation rate | $\text{mm day}^{-1}$ | 0.01 | 1000 |
| percexp | percolation exponent | - | 1 | 20 |
| sacpmlt | SAC model percolation multiplier for dry soil layer | - | 1 | 250 |
| sacpexp | SAC model percolation exponent for dry soil layer | - | 1 | 5 |
| iflwrte | interflow rate | $\text{mm day}^{-1}$ | 0.01 | 1000 |
| $\text{axv}_{\text{bexp}}$ | ARNO/VIC b exponent | - | 0.0001 | 3 |
| sareamax | maximum saturated area | - | 0.05 | 0.95 |
| loglamb | mean value of the topographic index | m | 5 | 10 |
| tishape | shape parameter for the topographic index Gamma distribution | - | 2 | 5 |
| $\text{qb}_{\text{powr}}$ | baseflow exponent | - | 1 | 10 |
| timedelay | time delay in runoff | days | 2.5 | 2.5 |

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
