# Peer review of "The impact of hydrological model structure on the simulation of extreme runoff events"

_Natural Hazards and Earth System Sciences, 2020_

## Referee Comment (RC1) · Anonymous Referee #1 · 4 Aug 2020

In this study the FUSE framework to unserstand model structural error is used to investigate the effects of model structure on extreme events in different climate zones. The authors do not use real catchments to investigate the model structural effects but a synthetic approach with a given range of parameter sets (the same for each all climate zones). The topic of investigating structural errors is very relevant and the application in different climatic regions is interesting. The manuscript is written clearly and follows a logical structure, even though not quite the classic one.

While I generally like the methodological approach I am not fully convinced in every aspect , which the authors might explain in more detail.

Main points: - The parameter ranges are taken from the original FUSE paper and applied in different climate zones. I am not convinced that the parameter space is fully

(or sufficiently) sampled using theses ranges. For very different regions than the ones where the models were intended and developed for the ranges might be quite different and a stop in increase of change using the Kolmogorov-Sirnoff test might not indicate that the space as sufficiently sampled, but could also be that there is a region of the parameter space that is not considered at all by the study set up.

- I am also not fully convinced that the very same parameter range should be applied for the catchments that can be found in different zones, hence I cannot understand why in the synthetic test these ranges should be the same and not a plausible range known from or tested in real catchments from these zones.

- How much do the additional snow routine parameters potentially influence the plausible parameter ranges of the other parameters? I would argue that that could change quite a bit and again would expect some kind of evaluation for instance by using real catchments from the respective regions.

- How much could using the same parameters in the snow routine effect the results? The very same degree-day was used despite the different climate zones. for snow influenced catchments the snow routine is crucial and varying for instance the degree-day will have large differences in the simulations. Please discuss.

- One of the objectives of the study is to link extreme event via their return periods to their sensitivity to model structure if the extreme events are simulated. The authors use daily data and daily simulation, however, often very large events occur at shorter time scales. How could the apporach be extended to these or would that shift the return periods very much? I assume that might be particularly relevant for arid zones.

- the extreme events were selected by using the minimum and maximum, for many studies on extreme values (particularly low flows) a moving average is used to avoid effects of oscillations etc. in these ranges. Maybe that would also solve some of the problems with the hard-coded threshold?

[Figure]

- extreme values are looked at only in terms of timing and maximum/minimum simulated streamflow. Other parts of the events might be interesing as well (event volume, deficit, duration etc.), while I see that that is not the focus of this study, I would appreciate a couple of words on these and how easy or difficult the proposed method could be extended to these characteristics.

Minor comments:

- the terms "drought" and "low flow" are not clearly distinguished. While one (drought) can lead to the other, low flow is a seasonal characteristic of the flow regime. Maybe use instead of simply drought the term "hydrological drought" but since the study is really about low flows, why not fully leave out the term drought?

- form: the results part is slightly mixed with discussion parts (referring to other studies). Then a synthesis follows and then, when the reader would expect conclusions, a new discussion part starts. While it is interesting in a way, I would propose to change the order. A reader that is looking only at specific parts can easily find them without having to go through the full paper. The discusson bits in the result part could together with the synthesis become the first part of a discussion before going into the discussion about limitations of the study setup.

Line by line comments:

L2 Add "different" after "several"

L11 compared to?

L18 I would urge to better distinguish drought and low flow (effects on crop production are rather linked to meteorological drought)

L28 remove "a" before "statistical"

L29 does not need t be the GEV but a distribution that is suitable, please mention

L35 could refer also to the tails paper by Klemeš here (Klemeš, 2000)

L97 change the word "procedure"

L164 please add the ranges in a table since they are relevant for the study

L184 what does "mainly determined" mean here? A slight difference might be very important if coming from the model structure when using the same meteorological forcing

L211 will describe -> describes

L219 are -> is; what does "significantly sensitive" mean?

L227 Or none is adequate for this zone?

L245 indicates -> indicate

L288 that show deviant behaviour -> that deviate

L350 but there might be other formulations that were not tested, where that could show up

L357 there is no one hand...

L359 for shorter events also snow melt can become important

References

Klemeš, V., 2000. Tall tales about tails of hydrological distributions. I. Journal of Hydrologic Engineering, 5(3), pp.227-231.

---

## Author Comment (AC1) · 24 Aug 2020

We would like to thank the anonymous referee for their feedback. We appreciate that the reviewer generally liked the methodological approach. Below, we respond to the points raised by the reviewer.

Main points:

- The parameter ranges are taken from the original FUSE paper and applied in different climate zones. I am not convinced that the parameter space is fully (or sufficiently) sampled using theses ranges. For very different regions than the ones where the models were intended and developed for the ranges might be quite different and a stop in increase of change using the Kolmogorov-Smirnoff test might not indicate that the

space as sufficiently sampled, but could also be that there is a region of the parameter space that is not considered at all by the study set up.

We are not quite sure if we understand the point of the reviewer in this aspect. The upper and lower parameter boundaries are generally based on physical and conceptual understanding, and should in principle capture all values that these parameters could reasonably take, independent of climate or catchment type. As such, we do not doubt that the parameter ranges as provided by the FUSE paper are the right starting point for the sampling. Concerning the sampling itself; yes, given the high-dimensionality of the parameters and the relatively limited parameter sample size, there will be regions in parameter space that are unexplored. That is; there will be quite some space in between the samples. The latin hypercube sampling strategy, however, ensures that we sampled over the full parameter range and that there are no 'overlooked' regions or corners.

- I am also not fully convinced that the very same parameter range should be applied for the catchments that can be found in different zones, hence I cannot understand why in the synthetic test these ranges should be the same and not a plausible range known from or tested in real catchments from these zones

We believe that applying the same parameter range to different climates is well-justified. Most of the hydrologic model parameters are determined by catchment properties (landscape, geology, land use) and not by the climatic conditions. There can be a large variation of different catchment properties within the same climate zone, and therefore one can not beforehand limit or stretch the parameter range based on climate only. Of course, there are some relations between catchment properties and climate; elevation and/or slope can for instance influence climate but also catchment storage properties, vice-versa climate can influence the catchment through rain-induced erosion or through vegetation processes. However, this is difficult to predict or translate to generalities and depends on long soil formation processes and historical climate conditions.

- How much do the additional snow routine parameters potentially influence the plausible parameter ranges of the other parameters? I would argue that that could change quite a bit and again would expect some kind of evaluation for instance by using real catchments from the respective regions.

From a conceptual point of view, there is no reason to assume that snow routine parameters influence the parameter ranges of the other parameters. These ranges are determined independent of the snow process. Of course, when one would calibrate a model, it would make a difference for the final parameter values coming out of the calibration if snow parameters were included or not, but the parameter ranges of the other parameters would not be adapted for the calibration procedure.

- How much could using the same parameters in the snow routine effect the results? The very same degree-day was used despite the different climate zones. for snow influenced catchments the snow routine is crucial and varying for instance the degreeday will have large differences in the simulations. Please discuss

Degree day parameters not only depend on climate, but also on many local circumstances (such as the distribution over north and south facing slopes or wind conditions that are not necessarily specified in the Koppen-Geiger classification). As such, we think it is cleanest to keep the degree day parameters fixed across the different climates. It is a valid point, however, that the snow parameters were not sampled, whereas the other model parameters were. Also sampling the snow parameters would probably further broaden the uncertainty bands around the simulations. We will add a clarification and discussion of the treatment of snow parameters.

- One of the objectives of the study is to link extreme event via their return periods to their sensitivity to model structure if the extreme events are simulated. The authors use daily data and daily simulation, however, often very large events occur at shorter time scales. How could the approach be extended to these or would that shift the return periods very much? I assume that might be particularly relevant for arid zones.

[Figure]

Indeed in arid zones, extreme events are often related to flash floods which last for a few hours only. It would require higher temporal-resolution climate model output in order to be able to simulate such events. This would be computationally quite challenging, given also the localized and convective nature of the rainfall that triggers such flash floods. Our return-period method does allow for relatively easy translation from daily to hourly, but we are limited here by the possibilities on the climate modelling side. Currently, we implicitly assume that the 24h mean would also be among the highest if a flash flood occurred within those 24 hours. This is of course not necessarily the case, we will add a note on this to the discussion.

- The extreme events were selected by using the minimum and maximum, for many studies on extreme values (particularly low flows) a moving average is used to avoid effects of oscillations etc. in these ranges. Maybe that would also solve some of the problems with the hard-coded threshold?

Thank you for this suggestion. A moving average is indeed an option that we will investigate in order to see if this increases the robustness of the results. We expect, however, that it might not completely resolve the hard-coded threshold issue, since these periods are rather persistent.

- Extreme values are looked at only in terms of timing and maximum/minimum simulated streamflow. Other parts of the events might be interesting as well (event volume, deficit, duration etc.), while I see that that is not the focus of this study, I would appreciate a couple of words on these and how easy or difficult the proposed method could be extended to these characteristics.

Thank you for this suggestion. Indeed, extremes can be defined in many different ways, and min/max discharge or timing are only two of many. Event volume is generally a bit more challenging because it requires the definition of a start and an end of the event - equations exist for this but the parameters of these equations might be climate / catchment dependent. We will investigate this suggestion.

[Figure]

Minor comments

- The terms "drought" and "low flow" are not clearly distinguished. While one (drought) can lead to the other, low flow is a seasonal characteristic of the flow regime. Maybe use instead of simply drought the term "hydrological drought" but since the study is really about low flows, why not fully leave out the term drought?

We agree with the reviewer that the terms were used interchangeably. Indeed in the formal definition, drought is used for anomalies while low flows are a seasonal characteristic. Since we are looking for the most extreme low flows (basically negative anomalies in low flows) this could again be perceived as a drought. But to not further complicate the text, we will replace drought with low flow throughout the manuscript.

- Form: the results part is slightly mixed with discussion parts (referring to other studies). Then a synthesis follows and then, when the reader would expect conclusions, a new discussion part starts. While it is interesting in a way, I would propose to change the order. A reader that is looking only at specific parts can easily find them without having to go through the full paper. The discussion bits in the result part could together with the synthesis become the first part of a discussion before going into the discussion about limitations of the study setup.

Thank you for the suggestion. We will adapt the structure accordingly.

Line by line comments

All textual suggestions will be incorporated.

---

## Referee Comment (RC2) · Anonymous Referee #2 · 29 Sep 2020

Review of *The impact of hydrological model structure on the simulation of extreme runoff events* by Van Kempen et al.

The manuscript of Van Kempen et al. deals with the influence of model structures on the magnitude and timing of extreme events. To do so, the FUSE framework was used with ten model structures and 100 parameter sets. The models were applied for four different climate zones and forced with a simulated timeseries of 2000 years. The authors show that alterations in percolation and evaporation affect mostly the magnitude of high flow events, especially for the cold and temperate climate zones. For low flows, especially the lower and upper formulation mattered. Generally, the model structural uncertainty was found to be higher for the low flow situations. In the arid and tropical climate zones, almost all model simulations agreed on the timing of the events, which showed a reduced influence of the model structure.

Generally, I like how the authors approach the problem and believe the article is clearly written and to-the-point. It is relatively short, but concise. Nevertheless, there are several issues that the authors may need to address.

First, I am not sure if the parameter sampling strategy is sufficient. A sample size of 100 parameters is, in my view, extremely low. I like how the authors use a K-S-test to assess whether the sampled distribution differs from a benchmark set, and believe also that this could be a good approach to determine the appropriate number of samples. However, the benchmark sample size is also just 500 samples, which is also still relatively low. With eleven parameters, this means that the sampling density (defined as $N^{(1/p)}$, with p the number of dimensions and N the sample size), is just around 1.76. In other words, on average, there are less than two samples per parameter. I think this sample size should be increased to at least a couple of thousand, then the KS-test makes more sense and can be used to select a lower, proper number of samples for the rest of the analysis. Of course, I fully understand that there will be a computational burden to it, but the authors could do this also for a shorter time period as the 34 years used now in order to save resources.

The authors are also quite critical on their own results regarding the low flow events, which is a very good thing in itself. However, if there are indeed so many numerical artefacts here, and we can not fully trust the results, it may just be better to completely leave this analysis out and focus on the high flow analysis.

I also wonder how much the cell-based approach matters. Especially regarding floods, the size of the catchments matters, as the flood-wave will be routed through the river-network. There was no routing model included, so how much will this make a difference in the results? Or, in other words, are the cell sizes small enough to ignore the routing effects?

Lastly, it is not fully clear to me how the analysis on the timing of the extreme events works. Why do the resulting bar charts in Figure 7 have a varying number of events on the x-axis? Do these correspond with different parameters, model structures or different return periods?

To conclude, the manuscript is very promising and interesting. I really like the methodology, and think the article is well written. I hope the authors find my comments useful and I look forward to an improved version of the manuscript.

**Minor comments**
P2L28. on a statistical models → on a statistical model

P2.L34. Statistical model → statistical models?

P4.L96. I am getting a bit confused here, are you setting up the models for four selected grid cells? Can you provide some details?

P5.L126. Selected structures → selected model structures

P7.L151-152. The Kolmogorov-Smirnov...is significant. → This is a bit unclear, but I think you mean that you asses whether an empirical distribution based on one parameter sample is statistically different from another empirical distribution based on another parameter sample, correct?

P9.L189. A high value in the red row →  Maybe define also what this value is in the text.

P9.L191-192. I assume you also repeat this for each parameter set for each model structure, correct?

P10.L203-204. Why just four of them? I think you should also show the others, at least in the Supplementary Material.

P10.L207 as displayed in Figure 6 → Please help the reader a bit, how do I see this exactly?

P10.L211-214. This section...Figure 3d. → This sounds as if this text was originally below the header of section 3.1 and later moved here.

P10.L221. Has least impact → has the least impact

P12.L255. Soil moisture → The upper layer soil moisture is actually quite critical in extreme events, as with saturated soils more overland flow occurs.

P14.L320. Which are .. stability → why not check this with the developers?

P14.L.325. Our results...these events. → Can you elaborate and clarify how I can see this? I am not sure how the see this from the results as shown so far.

Fig3. The unit of probability seems strange to me. Is that correct? Please also add a legend.

Fig6. Please add what you mean with the lower limits in the caption, and an x-label with the return periods.

---

## Author Comment (AC2) · 1 Oct 2020

**Response to interactive comment of Anonymous Referee #2**

We would like to thank Anonymous Referee #2 for their constructive feedback. The reviewer indicated that the manuscript is promising and interesting, and well written. With the suggested feedback, we expect to further improve the manuscript.

*Generally, I like how the authors approach the problem and believe the article is clearly written and to- the-point. It is relatively short, but concise. Nevertheless, there are several issues that the authors may need to address.*

*First, I am not sure if the parameter sampling strategy is sufficient. A sample size of 100 parameters is, in my view, extremely low. I like how the authors use a K-S-test to assess whether the sampled distribution differs from a benchmark set, and believe also that this could be a good approach to determine the appropriate number of samples. However, the benchmark sample size is also just 500 samples, which is also still relatively low. With eleven parameters, this means that the sampling density (defined as N^(1/p), with p the number of dimensions and N the sample size), is just around 1.76. In other words, on average, there are less than two samples per parameter. I think this sample size should be increased to at least a couple of thousand, then the KS-test makes more sense and can be used to select a lower, proper number of samples for the rest of the analysis. Of course, I fully understand that there will be a computational burden to it, but the authors could do this also for a shorter time period as the 34 years used now in order to save resources.*

We agree that the parameter sampling is rather coarse, indeed because of computational constraints. Testing for shorter time periods however, has the disadvantage that we then cannot test the effect of parameters on the kind of events we are interested in (extreme events with long return periods, 34 years is already relatively short for that). Furthermore, we would like to emphasize that we used a Latin Hypercube Sampling Strategy, this means that for a sample size of 100, each parameter has 100 different values because the parameters are all sampled at the same time (this can be done under the assumption that the parameters are independent).

Based on the feedback of the reviewer, we have increased our benchmark sample size to 5000 (this used to be 500). The results are shown in Figure 1. We still observe that the D-statistic starts to stabilize at around 100 parameter samples, therefore we do think we can safely assume that a sample size of 100 is a reasonable size to capture variability introduced by parameters. This number seems smaller than found in many other studies, and probably relates to our variable of interest - only the maximum and minimum discharge.

[Figure]

Figure 1. Overview of the D-statistic for Qmax (upper panel) and Qmin (lower panel) across the four climates. De D-statistic is obtained benchmarked against a sample of 5000 parameter sets.

*The authors are also quite critical on their own results regarding the low flow events, which is a very good thing in itself. However, if there are indeed so many numerical artefacts here, and we can not fully trust the results, it may just be better to completely leave this analysis out and focus on the high flow analysis.*

We have considered focussing only on high flows based on the results, but in the end made a deliberate choice to include the low flow results as well, to overcome the so-called "publication bias" where only positive results are published. We hope to avoid people repeating the same study for low flows and finding the same problems, because we decided not to publish or include the problems arising when evaluating low flows. Furthermore, the negative results on the low flow events may guide further research efforts into improving the modelling of such flows.

*I also wonder how much the cell-based approach matters. Especially regarding floods, the size of the catchments matters, as the flood-wave will be routed through the river-network. There was no routing model included, so how much will this make a difference*

*in the results? Or, in other words, are the cell sizes small enough to ignore the routing effects?*

Indeed when applied to a specific catchment, the catchment size and the temporal resolution will determine whether routing can be ignored or not. Routing will delay the peak and decrease the peak height. We did not consider routing because we apply a synthetic experiment and therefore the routing parameters cannot be calibrated on a catchment outlet measurement station. The effect of the routing parameters on the peak are known (delay and attenuation) and consistent among the different model structures if the same routing procedure is applied: the routing has no effect on the generated runoff itself. In this way, we keep the comparison clean. It is, however, a valid point raised by the reviewer and we will add it to the discussion.

*Lastly, it is not fully clear to me how the analysis on the timing of the extreme events works. Why do the resulting bar charts in Figure 7 have a varying number of events on the x-axis? Do these correspond with different parameters, model structures or different return periods?*

The timing analysis is indeed rather complex to explain, we will try to further improve the description of the procedure. To explain the numbers on the x-axis: Since we evaluate the timing of events with a 500-year return period and we have a simulation period of 2000 years, each simulation will have 4 of these extreme events. If all the different simulations (with combinations of different parameters and different model structures) agreed upon the timing of this extreme event, indeed only 4 events would be identified in total, and the x-axis would go to a max of 4 with 4 fully filled stacked bar charts (indicated as the "theoretical max"). The number on the x-axis indicates the number of extreme events with a different timing. So, if the x-axis goes up to 20, it means that across all the simulations, 20 different 500-yr return period events with a different timing can be found. The higher the number on the x-axis, the more variation there is among the different simulations in the timing of 500yr-return period events. The height of the bar chart indicates how many simulations identified a particular event. In the temperate climate, for instance, 1 event is identified by all simulations because it has a fully coloured bar chart. However, there is large disagreement about the timing of the other 3 events given that 38 events with different timing were identified.

*To conclude, the manuscript is very promising and interesting. I really like the methodology, and think the article is well written. I hope the authors find my comments useful and I look forward to an improved version of the manuscript.*

Thank you!

***Minor comments***

Suggestions for textual corrections and textual additions for clarification will all be implemented.

*P7.L151-152. The Kolmogorov-Smirnov...is significant. → This is a bit unclear, but I think you mean that you asses whether an empirical distribution based on one parameter sample is statistically different from another empirical distribution based on another parameter sample, correct?*

That is correct, we will reformulate.

*P9.L191-192. I assume you also repeat this for each parameter set for each model structure, correct?*

Step b is the evaluation at the parameter set level, and step c is for each model structure. In the end, indeed, both parameter set and model structure are accounted for in this analysis. In response to this comment and the comment on Figure 7, we will further elaborate on this approach.

*P10.L203-204. Why just four of them? I think you should also show the others, at least in the Supplementary Material.*

We selected these four models to demonstrate and explain the analysis (that's the goal of this figure) - we thought that including all the models would make it less clear. The results for all the models are shown in Figure 6. We will test whether we can include all models in Figure 5 while still keeping things clear, e.g. by showing all other models in grey.

*P14.L320. Which are .. stability → why not check this with the developers?*

This is a good suggestion, we will inquire with the developer (Martyn Clark, and Nans Addor which is currently maintaining the FUSE code).

*P14.L.325. Our results...these events. → Can you elaborate and clarify how I can see this? I am not sure how the see this from the results as shown so far.*

This is indeed not indicated very clearly in the previous parts of the results section. We will evaluate our statement and indicate references to this conclusion in the result section where applicable/appropriate.

*Fig3. The unit of probability seems strange to me. Is that correct? Please also add a legend.*

It would be more correct indeed if the y-axis would be labelled as probability density rather than probability. Since the area under a probability density plot needs to be equal to one, the units of probability density are the reciprocal of the units of the x-axis, and thus correctly indicated in the plot. A legend will be added, thanks for the good suggestions.

---

## Author Response (AR1)

[revised manuscript text omitted]

**Response to Reviewers**

Dear editor,

Thank you for organising the review process. Both reviewers consider this study relevant and interesting, but also provided useful suggestions for further improvements. The main adaptations to the manuscript are:

- A different treatment of the hard-coded lower limits that our low-flow simulations touched upon. This was not a direct suggestion from the reviewers, but it was inspired by the feedback from the reviewers and relates to several points raised by the reviewers.

- Further clarified the procedure of the timing analysis

- Important considerations were added to the discussion, related to flash flood events, snow melt processes, and the evaluated extreme event characteristics.

Please find below a point-by-point discussion, where our answers to the reviewers are indicated in blue.

Yours sincerely,

Gijs van Kempen
Lieke Melsen
Karin van der Wiel

**Reviewer 1**

**Summary**

In this study the FUSE framework to unserstand model structural error is used to investigate the effects of model structure on extreme events in different climate zones. The authors do not use real catchments to investigate the model structural effects but a synthetic approach with a given range of parameter sets (the same for each all climate zones). The topic of investigating structural errors is very relevant and the application in different climatic regions is interesting. The manuscript is written clearly and follows a logical structure, even though not quite the classic one. While I generally like the methodological approach I am not fully convinced in every aspect , which the authors might explain in more detail.

We would like to thank the reviewer for the careful evaluation of our manuscript.

**Main points**

The parameter ranges are taken from the original FUSE paper and applied in different climate zones. I am not convinced that the parameter space is fully (or sufficiently) sampled using theses ranges. For very different regions than the ones where the models were intended and developed for the ranges might be quite different and a stop in increase of change using the Kolmogorov-Smirnoff test might not indicate that the space as sufficiently sampled, but could also be that there is a region of the parameter space that is not considered at all by the study set up.

We are not quite sure if we understand the point of the reviewer in this aspect. The upper and lower parameter boundaries are generally based on physical and conceptual understanding, and should in principle capture all values that these parameters could reasonably take, independent of climate or catchment type. As such, we do not doubt that the parameter ranges as provided by the FUSE paper are the right starting point for the sampling.

Concerning the sampling itself; yes, given the high-dimensionality of the parameters and the relatively limited parameter sample size, there will be regions in parameter space that are unexplored. That is, there will be quite some space in between the samples. The Latin Hypercube Sampling (LHS) strategy, however, ensures that we sampled over the full parameter range and that there are no 'overlooked' regions or corners - in other words, with our sample size of 100 using an LHS sampling strategy, for each parameter 100 different values are tested. Besides, in response to the feedback of Reviewer 2, we have increased the benchmark against which the sample is tested to 5000 samples. There is still convergence around 100 samples, which implies that most of the variability is already captured when 100 samples are taken, and only marginal increases in variability can be expected when the sample size is further increased - at the expense of a lot of computer power. As such, we believe that we took a valid approach.

I am also not fully convinced that the very same parameter range should be applied for the catchments that can be found in different zones, hence I cannot understand why in the synthetic test these ranges should be the same and not a plausible range known from or tested in real catchments from these zones.

We believe that applying the same parameter range to different climates is well-justified. Most of the hydrologic model parameters are determined by catchment properties such as landscape, geology, and land use, that determine for instance storage capacity. There can be a large variation of different catchment properties within the same climate zone, and therefore one can not beforehand limit or stretch the parameter range based on climate only.

Of course, there are some relations between catchment properties and climate; elevation and/or slope can for instance influence climate but also catchment storage properties, vice-versa climate can influence the catchment through rain-induced erosion or through vegetation processes. However, this is difficult to predict or translate to generalities and depends on long soil formation processes and historical climate conditions. It is as such not straight forward to substantially limit the parameter range of hydrological models given a certain climate. We are also not aware of any such endeavours or methods in the scientific literature.

How much do the additional snow routine parameters potentially influence the plausible parameter ranges of the other parameters? I would argue that that could change quite a bit and again would expect some kind of evaluation for instance by using real catchments from the respective regions.

From a conceptual point of view, there is no reason to assume that snow routine parameters influence the parameter ranges of the other parameters. These ranges are determined independent of the snow process. Of course, when one would calibrate a model, it would make a difference for the final parameter values coming out of the calibration if snow parameters were included or not, but the parameter ranges of the other parameters would not be adapted for the calibration procedure.

How much could using the same parameters in the snow routine effect the results? The very same degree-day was used despite the different climate zones. for snow influenced catchments the snow routine is crucial and varying for instance the degree day will have large differences in the simulations. Please discuss

Since we only use one snow formulation (the degree day method), the snow processes are not a central part of this study; for all other processes, we use several formulations. For a fair comparison, we think it is cleanest to keep the snow parameters fixed and consider this a pre-processing part. Also sampling the snow parameters would probably further broaden the uncertainty bands around the simulations.

It is true that in some climates, extreme events might be influenced by snow, and we currently do not account for that. We have added and clarified this point in the discussion (line numbers 378-382).

One of the objectives of the study is to link extreme event via their return periods to their sensitivity to model structure if the extreme events are simulated. The authors use daily data and daily simulation, however, often very large events occur at shorter time scales. How could the approach be extended to these or would that shift the return periods very much? I assume that might be particularly relevant for arid zones.

Indeed in arid zones, extreme events are often related to flash floods which last for a few hours only. It would require higher temporal-resolution climate model output in order to be able to simulate such events. This would be computationally quite challenging, given also the localized and convective nature of the rainfall that triggers such flash floods. Our return-period method does allow for relatively easy translation from daily to hourly, but we are limited here by the possibilities on the climate modelling side. Currently, we implicitly assume that the 24h mean would also be among the highest if a flash flood occurred within those 24 hours. This is of course not necessarily the case. We thank the reviewer for this valuable suggestion and added a discussion on short extreme events, as being particularly relevant in arid climates (line numbers 365-368).

The extreme events were selected by using the minimum and maximum, for many studies on extreme values (particularly low flows) a moving average is used to avoid effects of oscillations etc. in these ranges. Maybe that would also solve some of the problems with the hard-coded threshold?

We would like to thank the reviewer for this valuable suggestion. Indeed, using a moving average is not uncommon for evaluating low flows. We have checked our results and evaluated the impact of using moving averages of up to 7 days for the minimum flow. However, since we are looking at quite extreme events, in all cases the lowest low flows persisted longer than 7 days, indicating that using a moving average did not make any difference to the results.

We have, however, decided to interpret the hard-coded lower limits in a different manner. The lower limits themselves might be numerical artefacts, but conceptually, these lower limits indicate that the river has run dry. For every model run, we have evaluated what the hard-coded lower limit was (this differed per model structure), and set this equal to 0. As such, we no longer find significant differences between two models if both models reach their hard-coded lower limit, since they were both set to 0 and both indicate that the river falls dry. We think this is conceptually much stronger and it puts less emphasis on numerical artefacts.

Extreme values are looked at only in terms of timing and maximum/minimum simulated streamflow. Other parts of the events might be interesting as well (event volume, deficit, duration etc.), while I see that that is not the focus of this study, I would appreciate a couple of words on these and how easy or difficult the proposed method could be extended to these characteristics.

We agree with the reviewer that max and min flow are only two of many relevant signatures of hydrological extremes. We have added a section to the discussion, where we discuss several other signatures that could be investigated in the same fashion (line numbers 422-430).

**Minor comments**
The terms "drought" and "low flow" are not clearly distinguished. While one (drought) can lead to the other, low flow is a seasonal characteristic of the flow regime. Maybe use instead of simply drought the term "hydrological drought" but since the study is really about low flows, why not fully leave out the term drought?

We agree with the reviewer that the terms were used interchangeably and that this could cause unnecessary confusion. We have replaced all instances of 'drought' to low flows (expect for the first sentence).

Form: the results part is slightly mixed with discussion parts (referring to other studies). Then a synthesis follows and then, when the reader would expect conclusions, a new discussion part starts. While it is interesting in a way, I would propose to change the order. A reader that is looking only at specific parts can easily find them without having to go through the full paper. The discussion bits in the result part could together with the synthesis become the first part of a discussion before going into the discussion about limitations of the study setup.

We thank the reviewer for this suggestion. We have restructured the manuscript by moving the synthesis part to the discussion section.

All minor textual suggestions have been implemented and addressed.

**Reviewer 2**

The manuscript of Van Kempen et al. deals with the influence of model structures on the magnitude and timing of extreme events. To do so, the FUSE framework was used with ten model structures and 100 parameter sets. The models were applied for four different climate zones and forced with a simulated timeseries of 2000 years. The authors show that alterations in percolation and evaporation affect mostly the magnitude of high flow events, especially for the cold and temperate climate zones. For low flows, especially the lower and upper formulation mattered. Generally, the model structural uncertainty was found to be higher for the low flow situations. In the arid and tropical climate zones, almost all model simulations agreed on the timing of the events, which showed a reduced influence of the model structure.

Generally, I like how the authors approach the problem and believe the article is clearly written and to-the-point. It is relatively short, but concise. Nevertheless, there are several issues that the authors may need to address.

We would like to thank the reviewer for taking the time to review our manuscript. We are happy to read that the reviewer appreciates our approach.

**Main points**

First, I am not sure if the parameter sampling strategy is sufficient. A sample size of 100 parameters is, in my view, extremely low. I like how the authors use a K-S-test to assess whether the sampled distribution differs from a benchmark set, and believe also that this could be a good approach to determine the appropriate number of samples. However, the benchmark sample size is also just 500 samples, which is also still relatively low. With eleven parameters, this means that the sampling density (defined as $N^{(1/p)}$, with p the number of dimensions and N the sample size), is just around 1.76. In other words, on average, there are less than two samples per parameter. I think this sample size should be increased to at least a couple of thousand, then the KS-test makes more sense and can be used to select a lower, proper number of samples for the rest of the analysis. Of course, I fully understand that there will be a computational burden to it, but the authors could do this also for a shorter time period as the 34 years used now in order to save resources.

We agree that the parameter sampling is rather coarse, indeed because of computational constraints. Testing for shorter time periods however, has the disadvantage that we then cannot test the effect of parameters on the kind of events we are interested in (extreme events with long return periods, 34 years is already relatively short for that). Furthermore, we would like to emphasize that we used a Latin Hypercube Sampling Strategy, this means that for a sample size of 100, each parameter has 100 different values because the parameters are all sampled at the same time (this can be done under the assumption that the parameters are independent).

Based on the feedback of the reviewer, we have increased our benchmark sample size to 5000 where this used to be 500 (see Figure 2 of the revised manuscript). We still observe that the D-statistic starts to stabilize at around 100 parameter samples, therefore we do think we can safely assume that a sample size of 100 is a reasonable size to capture variability introduced by parameters. This number seems smaller than found in many other studies, and probably relates to our variable of interest - only the maximum and minimum discharge.

The authors are also quite critical on their own results regarding the low flow events, which is a very good thing in itself. However, if there are indeed so many numerical artefacts here, and we can not fully trust the results, it may just be better to completely leave this analysis out and focus on the high flow analysis.

As indicated in our earlier response on this review, we have considered focusing only on high flows based on the results, but in the end made a deliberate choice to include the low flow results as well, to overcome the so-called "publication bias" where only positive results are published.

We did, however, re-evaluate the way we treat low flows and the numerical problems, and decided to take a different approach. The lower limits themselves might be numerical artefacts, but conceptually, these lower limits indicate that the river has run dry. For every model run, we have evaluated what the hard-coded lower limit was (this differed per model structure), and set this equal to 0. As such, we no longer find significant differences between two models if both models reach their hard-coded lower limit, since they were both set to 0 and both indicate that the river falls dry. We think this is conceptually much stronger and it puts less emphasis on numerical artefacts. As such, we feel more confident in presenting the low flow results.

I also wonder how much the cell-based approach matters. Especially regarding floods, the size of the catchments matters, as the flood-wave will be routed through the river-network. There was no routing model included, so how much will this make a difference in the results? Or, in other words, are the cell sizes small enough to ignore the routing effects?

Our text was confusing considering the routing. We did apply a simple routing scheme, but kept the scheme and the parameters fixed. Indeed when applied to a specific catchment, the catchment size and the temporal resolution will determine whether routing can be ignored or not. The effect of the routing parameters on the peak are known, namely delay and attenuation, and consistent among the different model structures if

the same routing procedure is applied. The routing has no effect on the generated runoff itself. Therefore, we decided to not to sample the routing parameters.

This can most clearly be explained for the high flows which we evaluate at max peak discharge: the routing parameter that decreases the peak height (by increasing diffusion) would dominate the results. Therefore, all other signals related to the underlying processes get lost, while all that the routing does is redistributing the runoff over time.

In the non-synthetic case, the routing parameters can be calibrated to a discharge outlet, but this is not the case for our synthetic study. Sampling the routing would lead to a result already known beforehand; the parameter that leads to lowest diffusion leads to highest peaks, but this does not provide any insights in the underlying processes. We have added an explanation to the main text (line numbers 141-145).

Lastly, it is not fully clear to me how the analysis on the timing of the extreme events works. Why do the resulting bar charts in Figure 7 have a varying number of events on the x-axis? Do these correspond with different parameters, model structures or different return periods?

The timing analysis is indeed rather complex. To explain the numbers on the x-axis: Since we evaluate the timing of events with a 500-year return period and we have a simulation period of 2000 years, each simulation will have 4 of these extreme events. If all the different simulations (with combinations of different parameters and different model structures) agreed upon the timing of this extreme event, indeed only 4 events would be identified in total, and the x-axis would go to a max of 4 with 4 fully filled stacked bar charts (indicated as the "theoretical max"). The number on the x-axis indicates the number of extreme events with a different timing. So, if the x-axis goes up to 20, it means that across all the simulations, 20 different 500-yr return period events with a different timing can be found. The higher the number on the x-axis, the more variation there is among the different simulations in the timing of 500yr-return period events. The height of the bar chart indicates how many simulations identified a particular event. In the temperate climate, for instance, 1 event is identified by all simulations because it has a fully coloured bar chart. However, there is large disagreement about the timing of the other 3 events given that 38 events with different timing were identified. We have elaborated on the explanation of this procedure (line numbers 200-232). We hope the explanation is now clear.

To conclude, the manuscript is very promising and interesting. I really like the methodology, and think the article is well written. I hope the authors find my comments useful and I look forward to an improved version of the manuscript.

Thank you!

All minor textual suggestions and required clarifications have been implemented.

---

## Referee Report (RR1)

**Review of **The impact of hydrological model structure on the simulation of extreme runoff events** by Van Kempen et al.**

The study of Van Kempen et al. evaluates the role of model structure on the modelling of extreme events, and they concluded it varied for different climate zones. This revision of the paper shows many improvements, and I am happy that my previous comments were found useful. I am especially happy that the authors decided to increase the benchmark sample size to 5000, which should better reflect the true distribution. It is still not really high, but acceptable. Then, I still have some comments, that are all more text-related, and I believe relatively minor and easy to correct.

First, the explanation about the timing of events is a lot clearer now. However, there are still a few confusing issues here. If I understood correctly now, each bar in Figure 7 relates to an event. In that case, the x-label is quite confusing, as it implies a cumulative and active selecting of events when moving to the right. "Event number", or just "Events" would cover it better probably. The gray theoretical maximum also confuses here, as these are displayed at a high event number, whereas these four bars could be displayed at any location in the chart, correct?

The authors renamed one section now also to Discussion, which is good, but there are sometimes paragraphs in the results that are part of the discussion (for example P13.L285 and onwards). So I suggest to go over the manuscript once more, and make a more distinct separation of results and discussion elements, or merge it under Results and Discussion.

To conclude, I like the paper and hope these comments are useful again. I am looking forward to the final version of the manuscript.

**Minor comments**

Throughout the paper: significance is stated with p<0.05 and p>0.05, for full correctness, this should be  $p \le 0.05$  and p > 0.05.

P3.L.76. To capture the complete parameter space  $\rightarrow$  please rephrase, you don't capture the full parameter space with 100 samples, but sample in a smart way to *represent* the full parameter space. P6.L141-145. Please also mention the modelling resolution and cellsize, this matters especially regarding routing and peak flows.

P10.L238. As depicted in Figure 3a  $\rightarrow$  Maybe better to stress that 3a is merely and example showing the principle.

P11.L244-252. Please be specific when discussing Figure 6, as it shows all climate zones, whereas Figure 5 just shows the tropical climate zone.

P.12.L275. In most climate zones, formulations  $\rightarrow$  you mean changing the formulations, correct? P12.L279. What you describe here is not really a hydrological process.

P13.L285. From here it seems more a discussion. This is fine, but then call it Results and Discussion, but as there is now a section Discussion, I think you may need to move these paragraphs there too. P13.L285-287. This is a repetition of the results presented before.

P14.L304-305. The magnitudes...formulations.  $\rightarrow$  What do you mean exactly? How are they similar? P14.L310. This depends also on the scale that you are modelling. In a small catchment, it is relevant, in a really big catchment, it is not, as it may re-infiltrate.

Caption Fig4. The coloured...as shown in panel (c).  $\rightarrow$  I am still confused, don't you mean that each row in (c) represents one color in the bar of (d)? Instead of just the blue row?

Fig5. It looks really nice, but is it correct that the figure labels (a)-(d) all have a different color?

---

## Author Response (AR2)

**The impact of hydrological model structure on the simulation of extreme runoff events**
**Response to Reviewers**

Dear editor,

Thank you for organising the review process once again. Both reviewers recognise the improvement of the manuscript after the major revisions and are happy to see that their comments were taken seriously. In the new reviews, they provided useful text-related comments, which were corrected in this new manuscript. The main adaptations to the manuscript are:

- Summarizing parts of the Results section were moved to the Discussion section

- The same format was used for all figures (labels and panels)

- All text-related comments were addressed

Please find below a point-by-point discussion, where our answers to the reviewers are indicated in blue.

Yours sincerely,

Gijs van Kempen
Lieke Melsen
Karin van der Wiel

**Reviewer 1**

The authors took great effort to implement the previous review comments and in my opinion the manuscript got much clearer. I have only some technical (mostly wording) comments to add that the authors might want to consider.

We would like to thank the reviewer once again for the careful evaluation of our revised manuscript.

**Text-related comments**

- L28 replace "frequently" by "most of the times"

- L123 please add a short summary how this choice was made by Kustas et al.

- L439 remove "now"

All text-related comments were changed and a short method description for the value of the degree-day factor was added (Lines 123-124 in the modified manuscript)

Figure 3 Change "Discharge" in x-axis of (d) to "Runoff" to keep the terminology consistent.

Thanks for this useful comment. We changed the figure accordingly, and therefore, the consistency of the terminology was improved.

Figure 4 caption: remove "conducted"

We changed this in the revised manuscript as well.

We want to thank the reviewer for these text-related comments, which improved the manuscript.

**Reviewer 2**

The study of Van Kempen et al. evaluates the role of model structure on the modelling of extreme events, and they concluded it varied for different climate zones. This revision of the paper shows many improvements, and I am happy that my previous comments were found useful. I am especially happy that the authors decided to increase the benchmark sample size to 5000, which should better reflect the true distribution. It is still not really high, but acceptable. Then, I still have some comments, that are all more text-related, and I believe relatively minor and easy to correct.

We would like to thank the reviewer for taking the time to review our revised manuscript. We are happy to read that the reviewer appreciates our effort and recognises the improvement. We agree that increasing the benchmark leads to a better reflection of the true distribution.

**Main points**

First, the explanation about the timing of events is a lot clearer now. However, there are still a few confusing issues here. If I understood correctly now, each bar in Figure 7 relates to an event. In that case, the x-label is quite confusing, as it implies a cumulative and active selecting of events when moving to the right. "Event number", or just "Events" would cover it better probably. The gray theoretical maximum also confuses here, as these are displayed at a high event number, whereas these four bars could be displayed at any location in the chart, correct?

Thanks for this suggestion, we changed the x-label of both Figure 4d and 7 to "Event number". The theoretical maximum is clarified in lines 233-234 in the modified manuscript, these lines explain that the theoretical maximum is always on the right side of the figure due to the fact that the events are sorted based on occurrence.

The authors renamed one section now also to Discussion, which is good, but there are sometimes paragraphs in the results that are part of the discussion (for example P13.L285 and onwards). So I suggest to go over the manuscript once more, and make a more distinct separation of results and discussion elements, or merge it under Results and Discussion.

We have moved the summarizing paragraphs in section 3.1 to the discussion section as suggested by the reviewer. Results and discussion are well separated now.

**Text-related comments**

Throughout the paper: significance is stated with $p < 0.05$ and $p > 0.05$, for full correctness, this should be $p \leq 0.05$ and $p > 0.05$.

That is correct, we changed this throughout the paper.

Caption Fig4. The coloured...as shown in panel (c). I am still confused, don't you mean that each row in (c) represents one color in the bar of (d)? Instead of just the blue row?

We clarified this in the caption of Figure 4, by further elaborating on the method.

Fig5. It looks really nice, but is it correct that the figure labels (a)-(d) all have a different color?

That is correct, the colours of the labels match the boxes in Fig 6, and help to guide the reader. An explanation of this is provided in the figure caption. If journal policy doesn't allow coloured labels, we will provide a figure version with black labels to the copy-editors.

All minor textual suggestions and required clarifications have been implemented.

To conclude, I like the paper and hope these comments are useful again. I am looking forward to the final version of the manuscript.

Thank you!